# The prevalence of psychiatric comorbidities in adult ADHD compared with non-ADHD populations: A systematic literature review

Won-Seok Choi[1◈], Young Sup Woo[2◈], Sheng-Min Wang[2], Hyun Kook Lim[2], Won-Myong Bahk[2]*

**1** Department of Psychiatry, Hallym University Sacred Heart Hospital, Anyang, Korea, **2** Department of Psychiatry, College of Medicine, The Catholic University of Korea, Seoul, Korea

◈ These authors contributed equally to this work.
* wmbahk@catholic.ac.kr

**Data Availability Statement:** All relevant data are within the paper and its Supporting Information files.

## Abstract

Comorbid psychiatric disorders in adults with ADHD are important because these comorbidities might complicate the diagnosis of ADHD and also worsen the prognosis. However, the prevalence of comorbid psychiatric disorders in adult ADHD varies according to the diagnostic tools used and the characteristics of target populations. The purpose of this review was to describe the prevalence of comorbid psychiatric disorders in adults with ADHD compared with adults without ADHD. Thirty-two studies published before August 2022 were identified and classified according to diagnosis of other psychiatric disorder in those with ADHD. The most frequent comorbid psychiatric disorder in the ADHD group was substance use disorder (SUD), followed by mood disorders, anxiety disorders, and personality disorders. The prevalence of these four disorders was higher in the ADHD group, whether or not subjects were diagnosed with other psychiatric disorders. In addition, the diversity of ADHD diagnostic tools was observed. This also might have affected the variability in prevalence of comorbidities. Standardization of ADHD diagnostic tools is necessary in the future.

## Introduction

ADHD(attention-deficit/hyperactivity disorder) is a common psychiatric disorder presenting persistent inattention and/or with hyperactivity/impulsivity [1], which is associated with considerable problems in personal, social, and occupational areas [2]. While ADHD is well studied in children, it is recently being studied in adults as well. According to a previous meta-analysis, 65% of children who were diagnosed with ADHD have persistent ADHD symptoms in adulthood [3]. In addition, the prevalence of ADHD in adults is known to reach 2.5% [4], which is moderate compared to its prevalence in children, which is about 5% [5].

Although comorbid psychiatric disorders are common in both adults and children, the comorbidity rate is higher in adults; as many as 80% of adults with ADHD are reported to have at least one comorbid psychiatric disorder [6–8]. In clinical adult ADHD samples, substance use disorder (SUD), mood disorder, anxiety disorder, and antisocial personality

**Funding:** The author(s) received no specific funding for this work.

**Competing interests:** The authors have declared that no competing interests exist.

disorder (ASPD) are the most common comorbid disorders [9, 10], and these mental disorders can adversely affect patient prognosis. Furthermore, research revealed that comorbid psychiatric disorders cause considerable functional impairment in individuals with ADHD and place a great burden on society [11].

For this reason, several cross-sectional studies have been conducted on various populations including clinical and general settings over 30 years to evaluate the prevalence of comorbid psychiatric conditions in adults with ADHD. However, the prevalence of comorbid psychiatric disorders varied according to characteristics of the subjects, including country, race, gender, and other socioeconomic characteristics as well as the screening or diagnostic tools applied [10, 12]. Moreover, since ADHD has been recognized in adults, diagnostic tools for adult ADHD and its comorbid disorders have changed over time [13], and the interest in clinical diagnoses and optimal treatments in adults with ADHD has also increased [14]. These factors might have contributed to the divergent prevalence rate of ADHD and comorbid disorders in adults.

However, to the best of our knowledge, despite the high prevalence of psychiatric disorders documented in previous studies in the adult ADHD subjects [10, 11, 15, 16] and their importance in the clinical field, no systematic literature review has specifically compared the prevalence of comorbid psychiatric disorders between adults with and without ADHD. Considering the high prevalence of adult ADHD and its impact on quality of life, a perspective on frequent comorbid psychiatric disorders would be helpful for individuals with ADHD and clinicians. Thus, the aim of our study was to ascertain the difference in the prevalence rates of comorbid psychiatric disorders between adults with and without ADHD including both clinical and general populations.

## Methods

### Study search and data sources

The methodology of the present review followed the Preferred Reporting Items for Systematic Reviews and Meta-Analysis (PRISMA) [17]. Following PRISMA guideline, we conducted research based on PICO (Population, Intervention, Comparison and Outcome). The target population was adults with diagnosed ADHD. We compared the prevalence of comorbid psychiatric disorders between ADHD and non-ADHD patients. We searched electronic libraries of PubMed, EMBASE, PsycINFO, PsycNET, and Google Scholar for publications regarding the epidemiology and prevalence rate of comorbidities of adult ADHD published from 1 January 1990 to 1 August 2022. The initial search was conducted by two authors (WSC, YSW) using the following terms: Prevalence AND (ADHD OR ADD OR Attention Deficit) AND Adult AND (comorbidity OR comorbid) in titles or abstracts. Each database was updated as appropriate when preparing this submission for publication. Full electronic search strategies are provided in the S1 Text.

### Study selection

First, articles obtained from the initial search were de-duplicated by EndNote 20. Then, inclusion/exclusion screening was performed by lead authors (WSC, YSW) based on exclusion criteria of non-relevant articles (e.g., did not focus on adult patients or did not include psychiatric comorbidity data), non-English articles, full text not available, abstract-only papers, and articles that were not peer-reviewed. We included all types of research except a systematic review or meta-analysis defined by title and method. The initial inclusion/exclusion review was based on titles and abstracts; if the relevance of the article was unclear, a full-text review was performed to determine the eligibility of each study. After this initial process, the full texts

of all included articles were retrieved to evaluate our detailed eligibility criteria. Articles were included in the study if they 1) used samples of adult populations aged 18 years or older, 2) defined clear ADHD and non-ADHD groups by clinically diagnoses or using any diagnostic criteria (e.g., DSM (Diagnostic and Statistical Manual of Mental Disorders)) or tools for screening/diagnosing ADHD in adults (e.g., ASRS (Adult ADHD Self-report Scale)), 3) defined the prevalence rate of comorbid psychiatric disorders using any diagnostic tools for each psychiatric disorder (e.g. SCID (Structured Clinical Interview for DSM-IV)), and 4) directly compared ADHD and non- ADHD groups using statistical analysis. Any discrepancy between the two lead authors during study selection was resolved through discussion, and other authors were consulted if necessary. The inclusion consistency between the two authors was 94.1% (32/34).

## Data collection process

Microsoft Excel was used to develop a data extraction spreadsheet, and all included full-text articles were reviewed by both researchers (WSC, YSW), who also conducted the initial data search and study selection process. The extracted data were reviewed for consistency, and any queries that arose were resolved by discussion among the researchers. The lead author decided whether to include/exclude data by reviewing the specific articles.

## Measurements

Because there are various methods for diagnosing ADHD and psychiatric disorders in adults, we extracted the following variables from the articles ultimately included: 1) data describing the study characteristics, such as year of publication, country, or study design; 2) data describing the target population, like sample size, age range or mean age/SD, or gender composition; 3) diagnostic tools for adult ADHD and comorbid psychiatric disorders, whether clinical diagnosis was performed, and the diagnostic criteria for ADHD/psychiatric comorbidity; 4) study results including the prevalence rate of ADHD in the target population, prevalence rate of each psychiatric comorbidity in each ADHD and non-ADHD group, and any statistically significant comparable variables including odd ratios(ORs) with 95% confidence intervals or chi-square ($\chi^2$) test variables.

## Classification of studies

Based on several studies targeting nation-wide psychiatric comorbidities [18–20], assuming that the prevalence of comorbid psychiatric disorders is higher in groups of psychiatric patients, we decided to divide the study populations into general population group studies and clinical group studies. A clinical group study was defined as one in which the study population included patients who had previously been clinically diagnosed with any psychiatric disorder or had visited/been admitted either voluntarily or involuntarily to a hospital for treatment. A general group study was defined as that in which the whole study population was not diagnosed with any psychiatric disorders before the start of each study.

In addition, considering the specificity that the prevalence of ADHD among incarcerated people was five to 10 times higher than that of the general population [21], and that the prevalence of comorbid psychiatric disorders among inmates was higher than that of the general population [22], we classified data of incarcerated patients separately from other population groups. The incarcerated group study was defined as a study that only included incarcerated participants.

## Results

In total, 1768 articles were identified by the search method described above, and 335 duplicates were removed. After the duplicates were excluded, an additional 1121 articles were excluded by screening titles and abstracts. The remaining 292 articles were read in full and included in the analysis if they met the inclusion criteria of our study. Based on our study criteria, 260 articles were excluded for reasons noted in Fig 1. Thus, 32 studies comparing the prevalence rates of comorbid psychiatric disorders between ADHD and non-ADHD adult subjects were selected for systematic review.

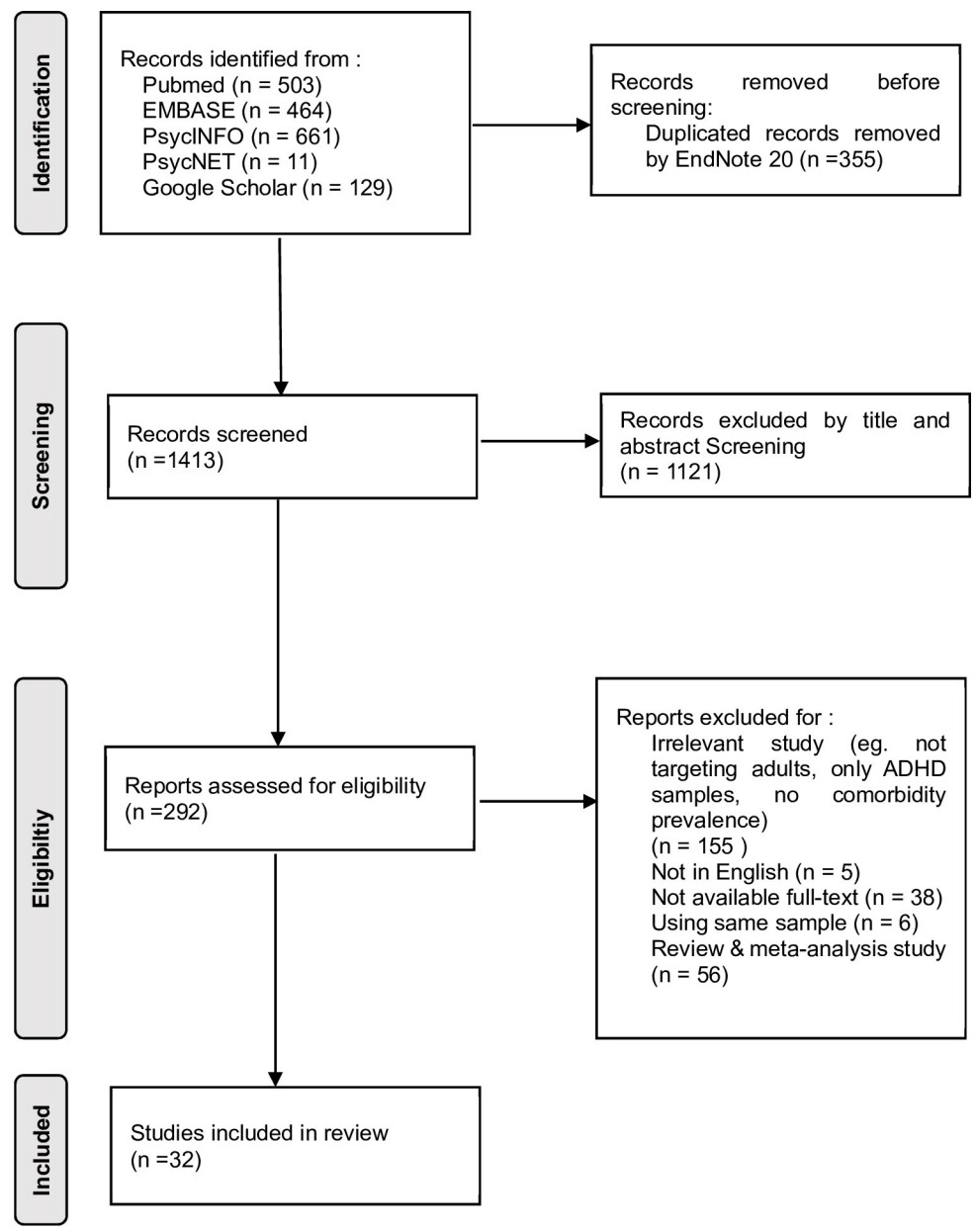

**Fig 1. PRISMA flow diagram.** Flow diagram of the manual screening process for eligible literature.

Of the 32 studies comparing the prevalence of comorbid psychiatric disorder between subjects with and without adult ADHD, according to our classification criteria, 11 studies involved general populations, 18 studies included psychiatric populations, and three studies focused on incarcerated populations. One of the three studies dealing with incarcerated populations involved only female inmates [23], and the other one study involved only male inmates [24].

## Diagnostic tools of included studies

In this review, 12 diagnostic tools including clinical diagnostic criteria like DSM or ICD(International Classification of Disease), were used to evaluate adult ADHD. In addition, five diagnostic tools were mainly used for comorbid psychiatric disorders. The most used evaluation tool for adult ADHD was Adult ADHD Self-Report Scale (ASRS) [25, 26], which was used in 12 studies. Five studies used the ASRS alone to evaluate ADHD in adults, and the rest of the studies used more than one tool to evaluate ADHD. The next most frequently used evaluation tools were Conner's Adult ADHD Diagnostic Interview for DSM-IV(CAADID) [27] and the Wender Utah Rating Scale (WURS) [28].

## Prevalence of mood disorders

Nineteen studies provided data comparing the prevalence of mood disorders (including depressive disorders and bipolar disorders) between ADHD patients and non-ADHD individuals [6, 16, 29–45]. In the general populations, the prevalence of any depressive disorder in the non-ADHD group was estimated at 1.2% [16] to 12.5% [35], compared to 8.6% [36] to 55% [6] in the ADHD group. In clinical populations, the prevalence of any depressive disorder in the non-ADHD group was estimated at 5.8% [32] to 39.6% [34], compared to 15.4% [32] to 39.7% [44] in the ADHD group. In the general population, the prevalence of any bipolar disorder in the non-ADHD group was estimated at 0.2% [35] to 3.6% [16] compared to 4.48% [42] to 35.3% [16] in the ADHD group. In clinical populations, the prevalence of any bipolar disorder in the non-ADHD group was estimated at 2.0% [37] to 19.5% [40], compared to 7.4% [34] to 80.0.% [29] in the ADHD group. There were no differences reported in the prevalence of mood disorders between ADHD and non-ADHD groups in the incarcerated population studies. Detailed information from each study is summarized in Table 1.

## Prevalence of anxiety and related disorders

Sixteen studies provided data comparing the prevalence of anxiety disorders including obsessive-compulsive disorder, somatoform disorders and trauma/stress-related disorders between ADHD patients and non-ADHD individuals [12, 16, 31, 34–36, 38, 39, 41–43, 45–49]. In general population, the prevalence of any anxiety disorders in the non-ADHD group was estimated at 0.5% [39] to 9.5% [36] compared to 4.3% [39] to 47.1% [36] in ADHD group. In clinical populations, the prevalence of any anxiety disorders in the non-ADHD group was estimated at 5.4% [46] to 40% [49] compared to 3.9% [34] to 84% [49] in the ADHD group. Only one study of incarcerated populations showed a difference in the prevalence of social phobia between non-ADHD and ADHD individuals [47]. Detailed information from each study is summarized in Table 2.

## Prevalence of substance use disorders and gambling disorder

Twenty-two studies provided data comparing the prevalence of substance use disorders (including addiction to alcohol, opioids, stimulants, cannabis, anxiolytics, and nicotine) and

Table 1. Studies comparing the prevalence of mood disorders between non-ADHD and ADHD subjects.

| Author | Year, | Country | N | % of male | Age | Assessment of ADHD | Assessment of comorbid psychiatric disorder | Design | Sample | Prev. of ADHD(%) (non-ADHD/ADHD) | Findings comparing non-ADHD and ADHD and prevalence of comorbid psychiatric disorders | non-ADHD, n (%) vs ADHD, n (%) |
|---|---|---|---|---|---|---|---|---|---|---|---|---|
| General sample | | | | | | | | | | | | |
| Solberg et al [43] | 2018 | Norway | 1,701,206 | 51.2% | 18≤ | ADHD medication at adult or ADHD diagnosis registered | ICD-10 | Cross-sectional study | General sample | 2.4% (40,103/ 1,661,103) | Bipolar Disorder | Women 13,183 (1.6%) vs 2,290 (12.9%) Men 9,009 (1.1%) vs 1,981 (8.9%) |
| | | | | | | | | | | | Major depressive disorder | Women 61,880 (7.6%) vs 5,138 (28.8%) Men 33,733 (4.0%) vs 4,516 (20.3%) |
| Chen et al [31] | 2018 | Norway | 5,551,807 | 50.81% | 18-64 | ICD-9: 314; ICD-10: F90 diagnosis | ICD | Cross-sectional study | General sample | 1.1% (61,129/ 5,490,678) | Depression | PR = 9.01 (8.92-9.10) |
| | | | | | | | | | | | Bipolar Disorder | PR = 19.96 (19.48-20.43) |
| Hesson and Fowler [35] | 2018 | Canada | 16,957 | NA | 20-64 | Self-report of ADHD (diagnosed by a health professional) | WHO-CIDI modified for the needs of CCHS-MH | Case-control study | General sample -national mental health survey | 2.9% (NA) | 12-month Major depressive disorder | 61 (12.5%) vs 113 (23.3%) $\chi2 = 59.94$ |
| | | | | | | | | | | | 12-month Bipolar disorder | 1 (0.2%) vs 20 (4.1%) $\chi2 = 17.73$ |
| Yoshimasu et al [16] | 2016 | US | 5,718 | NA | Mean age ADHD 30.2 (SD 1.9) Non-ADHD controls 30.2 (SD 2.0) | Childhood-identified ADHD with M.I. N.I (+) | M.I.N.I | Case-control study | General population- Birth cohort sample | NA, (68/ 335) | Hypomanic episode— current or past | 12 (3.6%) vs 24 (35.3%) OR adj 16.5 [7.2, 37.4] |
| | | | | | | | | | | | Dysthymia | 4 (1.2%) vs 11 (16.2%) OR adj 19.0 [5.4, 66.1] |
| | | | | | | | | | | | MDD | 9 (2.7%) vs 19 (27.9%) OR adj 15.2 [6.2, 37.4] |

(Continued)

**Table 1.** (Continued)

| Author | Year, | Country | N | % of male | Age | Assessment of ADHD | Assessment of comorbid psychiatric disorder | Design | Sample | Prev.of ADHD(%) (non-ADHD/ADHD) | Findings comparing non-ADHD and ADHD and prevalence of comorbid psychiatric disorders | non-ADHD, n (%) vs ADHD, n (%) |
|---|---|---|---|---|---|---|---|---|---|---|---|---|
| Park et al [39] | 2011 | South Korea | 6,081 | ADHD + 59.4% ADHD- 50.5% | 18–59 | ASRS-S v 1.1 (+) | K-CIDI (Korean Ver. of CIDI) | Epidemiological study | General sample | 1.1% (69/6,012) | Any mood disorder | 6.0% vs 27.1% OR 6.44 [3.70–11.19] |
| | | | | | | | | | | | Major depressive disorder | 5.5% vs 17.4% OR = 4.00 [2.10–7.63] |
| | | | | | | | | | | | Bipolar disorder | 0.2% vs 8.6% OR 29.94 [10.71–83.66] |
| Miller et al [38] | 2007 | US | 363 | 51.0% | 18–37 | K–SADS & structured interview | SCID-I, SCID-II | Case-control study | General sample - Recruited ADHD vs control group | NA, (152/211) | Mood disorder | NA, χ2 = 23.70 |
| Sobanski et al [6] | 2007 | Germany | 140 | 54.3% | Mean age ADHD+ 36.8 (SD 9.0) ADHD- 39.8 (SD 10.0) | WURS-K & BADDS | SCID-I | Case-control study | General sample -referred ADHD vs control group | NA, (70/70) | Affective disorders total | 18 (25.7%) vs 44 (60.7%) χ2 = 18.462 |
| | | | | | | | | | | | Major depressive episodes | 17 (24.3%) vs 40 (55%) χ2 = 15.010 |
| Kessler et al [36] | 2006 | US | 3,199 | NA | 18–44 | DIS-IV for childhood pathology & ACDS v 1.2 (ADHD-RS) | CIDI | Epidemiological study | General sample–national survey | 2.6% (NA) | Major depressive disorder | 7.8%vs 8.6% 4.2 OR 2.7 [1.5–4.9] |
| | | | | | | | | | | | Dysthymia | 1.9% vs 12.8% OR 7.5 [3.8–15] |
| | | | | | | | | | | | Bipolar | 3.1% vs 19.4% OR 7.5 [4.6–12.0] |
| | | | | | | | | | | | Any mood disorder | 11.1% vs 38.3% OR 5.0 [3.0–8.2] |
| Secnik et al [42] | 2005 | US | 4,504 | 64.3% | 18≤ | ICD-9 | ICD-9 | Case-control study | General sample–HPM database | (2,252/2,252) | Bipolar disorder | 0.58% vs 4.48% |
| | | | | | | | | | | | Depression | 2.93% vs 17.10% |

*(Continued)*

**Table 1.** (Continued)

| Author | Year | Country | N | % of male | Age | Assessment of ADHD | Assessment of comorbid psychiatric disorder | Design | Sample | Prev. of ADHD(%) (non-ADHD/ ADHD) | Findings comparing non-ADHD and ADHD and prevalence of comorbid psychiatric disorders | non-ADHD, n (%) vs ADHD, n (%) |
|---|---|---|---|---|---|---|---|---|---|---|---|---|
| **Clinical sample** | | | | | | | | | | | | |
| Woon and Zakaria [45] | 2019 | Malaysia | 120 | 94.2% | 18–65 | CAADID | M.I.N.I | Cross-sectional study | Psychiatric sample | 15.8% (101/19) | Manic/ hypomanic episode, lifetime | 8 (7.9%) vs 8 (42.1%) |
| Roncero et al [41] | 2019 | Spain | 726 | 72.5% | $18 \leq$ | ASRS ($14 \leq$) | DSM-IV-TR | Cross-sectional study | Psychiatric sample– treatment seeking AUD patients | 21.1% (573/153) | Mood disorder | 24.5% vs 49% $\chi^2 = 32.87$, OR 2.95 [2.2, 4.3] |
| Leung and Chan [37] | 2017 | Hong Kong | 254 | 28.7% | 18–64 | ASRS-v1.1 Symptom Checklist $\geq$17 & SDS $\geq$ 5 (Screening) + DIVA 2.0 (Diagnosis) | DSM-5 | Cross-sectional cohort study | Psychiatric sample– clinical outpatients | 19.3% (49/ 205) | Bipolar disorder | 2.0% vs 15.0% OR = 8.87 (1.83–42.9) (ADHD-combined type vs Non-ADHD) |
| Gorlin et al [34] | 2016 | US | 1,134 | 42% | Mean age 39.7 (SD 14.4) | DSM-IV based semi-structured clinical interview | SCID | Cross-sectional study | Psychiatric sample— clinical outpatients | 18.0% (204/903) | Major depressive disorder | 39.6% vs 29.4% OR = 0. 69 (0.49–0.96) |
| | | | | | | | | | | | Bipolar disorder | 3.4% vs 7.4% OR = 2.14 (1.09–4.02) |
| Fatséas et al [33] | 2016 | France | 217 | 66.4% | 18–65 | CAADID | DSM-IV for SUD SCIDII for BPD M.I.N.I. for others | Cross-sectional cohort study | Psychiatric sample– addiction clinical outpatients | 23.0% (50/ 167) | Current mood disorders | 36.8% vs 54.0% 0.030 |
| van Emmerik-van Oortmerssen et al [44] | 2014 | Australia, Belgium, France, Hungary, Netherlands, Norway, Spain, Sweden, Switzerland, US (IASP study) (37) | 1,205 | ADHD– 73.1% ADHD + 75.6% | 18–65 | CAADID | MINI Plus SCID-II | Cross-sectional study | Psychiatric sample— treatment-seeking SUD patients | 13.9% (168/ 1,037) | Current Depression— alcohol | 15.3% vs 39.7% OR 4.1 [2.1–7.8] |
| | | | | | | | | | | | Current (hypo) mania | 4.1% vs 14.9% OR 4.3 [2.1–8.7] |
| Duran et al [32] | 2014 | Turkey | 246 | NA | 18–60 | WURS score >36 & Turgay's Adult ADD/ADHD Evaluation Scale | SCID-I-CV, SCID-II | Cross-sectional study | Psychiatric sample— clinical outpatients | 15.9% (39/ 207) | Dysthymic Disorder | 12 (5.8%) vs 6 (15.4%) $\chi2 = 25.81$ |

*(Continued)*

**Table 1.** (Continued)

| Author | Year, | Country | N | % of male | Age | Assessment of ADHD | Assessment of comorbid psychiatric disorder | Design | Sample | Prev. of ADHD(%) (non-ADHD/ ADHD) | Findings comparing non-ADHD and ADHD and prevalence of comorbid psychiatric disorders | non-ADHD, n (%) vs ADHD, n (%) |
|---|---|---|---|---|---|---|---|---|---|---|---|---|
| Perugi et al [40] | 2013 | Italy | 96 | 59.4% | 18–65 | ASRS v 1.1 (+), & prior age 7 with ADHD sx | DSM-IV | Cross-sectional observation study | Psychiatric sample—Bipolar I, II disorder diagnosed | 19.8% (19/77) | BD I mixed state | 14 (18.2%) vs 10 (52.6%) χ2 = 9.6 |
| | | | | | | | | | | | BD I mania | 13 (16.9%) vs 0 (0%) χ2 = 3.7 |
| | | | | | | | | | | | BD I remission | 15 (19.5%) vs 0 (0%) χ2 = 0.1 |
| Ceraudo et al [29] | 2012 | Italy | 119 | 68.1% | Mean age ADHD+ 35.10 (SD 7.66) ADHD- 34.74 (SD 8.46) | ASRS-S v 1.1 (+) | DCTC (Diagnostic, Clinical and Therapeutic Checklist) | Cross-sectional study | Psychiatric sample – SUD outpatients | 18.35% (20/89) | Bipolar Disorder | 38 (43.2%) vs 16 (80.0%) χ2 = 8.84 |
| | | | | | | | | | | | Mixed/Manic | 15 (16.9%) vs 8 (40.0%) χ2 = 3.29 |
| Olsson et al [30] | 2022 | Sweden | 804 | 67.3% | 18≤ | ICD-10 or prescription of ADHD medication | ICD-10 | Cross-sectional study | Psychiatric sample– psychiatric emergency patient with NSSI | 11.6% (711/93) | Depression | 218 (31%) vs 12 (13%), χ2 = 12.7 |

OR: Odd Ratio, PR: Prevalence Ratio, NA: Not available (not identified in article)

SUD: Substance Use Disorder, AUD: Alcohol use disorder, BPD: Borderline Personality Disorder BD I: Bipolar I disorder, WHO-CIDI: World Health Organization version of the Composite International Diagnostic Interview, CCHS-MH: Community Health Survey–Mental Health, M.I.N.I: Mini-International Neuropsychiatric Interview, ASRS-S: Adult Self-Report Scale-Screener, ASRS: Adult Self-Report Scale, CIDI: Composite International Diagnostic Interview, K-SADS: Kiddie Schedule for Affective Disorders and Schizophrenia, SCID: Structured Clinical Interview for DSM-IV, SCID-I: Structured Clinical Interview for DSM-IV Axis I Disorders, SCID-I-CV: Structured Clinical Interview for DSM-IV Axis I Disorders, Clinician Version, SCID-II: Structured Clinical Interview for DSM-IV Axis II Disorders, WURS: Wender Utah Rating Scale, WURS-k: German short form of the Wender Utah rating scale, BADDS: Brown attention deficit disorder scale, DIS-IV: Diagnostic Interview Schedule for DSM-IV, ACDS: Adult ADHD Clinical Diagnostic Scale, ASHD-RS: ADHD Rating Scale, CAADID: Conner's Adult ADHD Diagnostic Interview for DSM-IV, ISAP: International ADHD in Substance use disorder Prevalence, NSSI: Nonsuicidal self-injury

**Table 2. Studies comparing the prevalence of anxiety and related disorder between non-ADHD and ADHD subjects.**

| Author | Year | Country | N | % of male | Age | Assessment of ADHD | Assessment of comorbid psychiatric disorder | Design | Sample | Prev. of ADHD (%) (non-ADHD/ADHD) | Findings comparing non-ADHD and ADHD and prevalence of comorbid psychiatric disorders | non-ADHD, n (%) vs ADHD, n (%) |
|---|---|---|---|---|---|---|---|---|---|---|---|---|
| **General sample** | | | | | | | | | | | | |
| Solberg et al [43] | 2018 | Norway | 1,701,206 | 51.2% | 18≤ | ADHD medication at adult or ADHD diagnosis registered | ICD-10 | Cross-sectional study | General sample | 2.4% (40,103/1,661,103) | Anxiety Disorders | Women 54,479 (6.7%) vs 4,676 (26.3%) Men 28,364 (3.3%) vs 4,054 (18.2%) |
| Chen et al [31] | 2018 | Norway | 5,551,807 | 50.81% | 18–64 | ICD-9: 314; ICD-10: F90 diagnosis | ICD | cross-sectional study | General sample | 1.1% (61,129/5,490,678) | Anxiety | PR = 9.12 (9.04–9.21) |
| Hesson and Fowler [35] | 2018 | Canada | 16,957 | NA | 20–64 | Self-report of ADHD (diagnosed by a health professional) | WHO-CIDI modified for the needs of CCHS-MH | Case-control study | General sample -national mental health survey | 2.9% (NA) | Generalized anxiety disorder | 15 (3.1%) vs 73 (15.1%) $\chi^2 = 42.30$ |
| Yoshimasu et al [16] | 2016 | US | 5,718 | NA | Mean age ADHD 30.2 (SD 1.9) Non-ADHD controls 30.2 (SD 2.0) | Childhood-identified ADHD with M.I.N.I (+) | M.I.N.I. | Case-control study | General population–Birth cohort sample | NA, (68/335) | PTSD | 3 (0.9%) vs 6 (8.8%) OR adj, 10.0 [2.9, 35.0] |
| | | | | | | | | | | | Social phobia-current | 4 (1.2%) vs 10 (14.7%) OR adj 12.8 [4.2, 39.4] |
| | | | | | | | | | | | OCD | 8 (2.4%) vs 14 (20.6%) OR adj 8.0 [3.3, 19.2] |
| | | | | | | | | | | | Generalized anxiety disorder | 30 (9.0%) vs 22 (32.4%) OR adj 4.7 [2.4, 9.0] |
| | | | | | | | | | | | Panic disorder–lifetime | 17 (5.1%) vs 9 (13.2) OR adj 2.6 [1.1, 6.2] |
| Park et al [39] | 2011 | South Korea | 6,081 | ADHD + 59.4% ADHD-50.5% | 18–59 | ASRS-S v 1.1 (+) | K-CIDI (Korean Ver. of CIDI) | Epidemiological study | General sample | 1.1% (69/6,012) | Any anxiety disorder | 6.3% vs 25.7% OR 5.46 [3.11–9.57] |
| | | | | | | | | | | | OCD | 0.6% vs 4.3% OR 8.26 [2.51–27.26] |
| | | | | | | | | | | | PTSD | 1.2% vs 7.2% OR 8.13[3.26–20.32] |
| | | | | | | | | | | | Social phobia | 0.5% vs 11.4% OR 7.57 [1.92–29.83] |
| | | | | | | | | | | | Specific phobia | 3.9% vs 11.4% OR 3.31 [1.52–7.18] |
| | | | | | | | | | | | Somatoform disorder | 1.1% vs 4.3% OR 4.30 [1.22–15.12] |

*(Continued)*

**Table 2.** (Continued)

| Author | Year | Country | N | % of male | Age | Assessment of ADHD | Assessment of comorbid psychiatric disorder | Design | Sample | Prev. of ADHD (%) (non-ADHD/ADHD) | Findings comparing non-ADHD and ADHD and prevalence of comorbid psychiatric disorders | non-ADHD, n (%) vs ADHD, n (%) |
|---|---|---|---|---|---|---|---|---|---|---|---|---|
| Miller et al [38] | 2007 | US | 363 | 51.0% | 18–37 | K–SADS & structured interview | SCID–I, SCID–II | Case-control study | General sample-Recruited ADHD vs control group | NA, (152/211) | Anxiety disorder | χ2 = 8.81 |
| Kessler et al [36] | 2006 | US | 3,199 | NA | 18–44 | DIS-IV for childhood pathology & ACDS v1.2 (ADHD-RS) | CIDI | Epidemiologic study | General sample-national survey | 2.6% (NA) | GAD | 2.6% vs 8.0% OR 3.2 [1.5–6.9] |
| | | | | | | | | | | | PTSD | 3.3% vs 11.9% OR 3.9 [2.1–7.3] |
| | | | | | | | | | | | Panic disorder | 3.1% vs 8.9% OR 3.0 [1.6–75.9] |
| | | | | | | | | | | | Agoraphobia | 0.7% vs 4.0% OR 5.5 [1.6–18.5] |
| | | | | | | | | | | | Specific phobia | 9.5% vs 22.7% OR 2.8 [1.7–4.6] |
| | | | | | | | | | | | Social Phobia | 7.8% vs 29.3% OR 4.9 [3.1–7.6] |
| | | | | | | | | | | | Any anxiety disorder | 19.5% vs 47.1% OR 3.7 [2.4–5.5] |
| Secnik et al [42] | 2005 | US | 4,504 | 64.3% | 18≤ | ICD-9 | ICD-9 | Case-control study | General sample–HPM database | NA (2,252/2,252) | Anxiety disorder | 3.46% vs 13.77% |
| **Clinical sample** | | | | | | | | | | | | |
| El Ayoubi et al [49] | 2020 | France | 551 | 83.8% | 18≤ | Both ASRS-S v1.1 (+) and WURS (26≤) | PCL-5 for PTSD | Cross-sectional study | Psychiatric inpatients with AUD | 19.8% (442/109) | PTSD | 179 (40%) vs 91 (84%) χ2 = 64.7 |
| Woon and Zakaria [45] | 2019 | Malaysia | 120 | 94.2% | 18–65 | CAADID | M.I.N.I | Cross-sectional study | Psychiatric sample -Forensic ward inpatient | 15.8% (101/19) | Generalized anxiety disorder | 20 (19.8%) vs 9 (47.4%) |
| Roncero et al [41] | 2019 | Spain | 726 | 72.5% | 18≤ | ASRS (14≤) | DSM-IV-TR | Cross-sectional study | Psychiatric patients–treatment seeking AUD patients | 21.1% (573/153) | Anxiety disorder | 10.5% vs 25.8% χ2 = 23.5 OR 2.95 [1.88, 4.64] |
| Reyes et al [48] | 2019 | US | 472 | 64.6% | 18–80 | PRISM | PRISM | Cross-sectional study | Psychiatric sample–inpatient & outpatient with DSM-IV-TR AUD diagnosis | 6.36% (30/442) | Anxiety disorders, current | 95 (21.5%) vs 14 (46.7%) |
| Gorlin et al [34] | 2016 | US | 1,134 | 42% | Mean age 39.7 (SD 14.4) | DSM-IV based semi-structured clinical interview | SCID | Cross-sectional study | Psychiatric sample-clinical outpatient | 18.0% (204/903) | Social phobia | 28.7% vs 38.2% OR = 1.46 (1.05–2.01) |
| | | | | | | | | | | | Any adjustment disorder | 9.4% vs 3.9% OR = 0.41 (0.18–0.82) |
| Retz et al [12] | 2016 | Germany | 163 | 86.5% | Mean age 40.2 (SD 9.4) | DSM-5 & WURS-k ≥ 30 | ICD-10 | Cross-sectional study | Psychiatric sample-GD dx according to ICD-10 | 25.2% (41-current ADHD/122) | Stress and adjustment disorders | 14 (8.6%) vs 7 (17.1%) χ2 = 5.70 |

(*Continued*)

**Table 2.** (Continued)

| Author | Year | Country | N | % of male | Age | Assessment of ADHD | Assessment of comorbid psychiatric disorder | Design | Sample | Prev. of ADHD (%) (non-ADHD/ADHD) | Findings comparing non-ADHD and ADHD and prevalence of comorbid psychiatric disorders | non-ADHD, n (%) vs ADHD, n (%) |
|---|---|---|---|---|---|---|---|---|---|---|---|---|
| Karahmet et al [46] | 2013 | Turkey | 90 | 53.3% | 18≤ | Turgay's Adult ADD/ADHD Evaluation Inventory & WURS | SCID-I | Cross-sectional study | Psychiatric sample-Bipolar disorder diagnosed | 23.3% (21/69) | OCD | 6 (10.7%) vs 4 (19.0%) |
| | | | | | | | | | | | Panic disorder | 3 (5.4%) vs 5 (23.8%) |
| Incarcerated sample | | | | | | | | | | | | |
| Moore et al [47] | 2016 | Australia | 88 | 76% | 18–72 | ASRS-S (+) & M.I.N.I plus (+) | M.I.N.I plus, PDQ-4, SCID-II | Cross-sectional study | Incarcerated sample | 17.0% (15/73) | Social phobia | 15.1% vs 46.7% OR = 4.39 [1.10, 17.56] |

OR: Odd Ratio, NA: Not available (not identified in article)

PTSD: Post-traumatic stress disorder, OCD: Obsessive-compulsive disorder, GAD: Generalized anxiety disorder, AUD: Alcohol used disorder, GD: Gambling disorder WHO-CIDI: World Health Organization version of the Composite International Diagnostic Interview, CCHS-MH: Community Health Survey–Mental Health, M.I.N.I: Mini-International Neuropsychiatric Interview, ASRS-S: Adult Self-Report Scale-Screener, ASRS: Adult Self-Report Scale, CIDI: Composite International Diagnostic Interview, K-SADS: Kiddie Schedule for Affective Disorders and Schizophrenia, SCID-I: Structured Clinical Interview for DSM-IV Axis I Disorders, SCID-II: Structured Clinical Interview for DSM-IV Axis II Disorders, WURS: Wender Utah Rating Scale, WURS-k: German short form of the Wender Utah rating scale, PRISM: Psychiatric research interview for substance and mental disorders, PDQ-4: Personality disorder diagnostic questionnaire for the DSM-IV

gambling disorders between ADHD and non-ADHD individuals [6, 12, 16, 23, 24, 31–33, 35–42, 45, 47, 48, 50–52]. In general populations, the prevalence of any substance use disorder in the non-ADHD group was estimated at 0% [6] to 16.6% [39] compared to 2.3% [35] to 41.2% [16] in the ADHD group. In clinical populations, the prevalence of any substance use disorder in the non-ADHD group was estimated to be 2.0% [45] to 72.2% [41] compared to 10.0% [48] to 82.9% [41] in the ADHD group. Two studies compared the prevalence of gambling disorder between ADHD and non-ADHD patients, and there was one study for each general/psychiatric population group, showing a statistically significant difference in prevalence [37, 39]. Two studies of incarcerated populations showed differences in the prevalence of benzodiazepine use disorder [47] and drug dependence [23]. Detailed information from each study is summarized in Table 3.

## Prevalence of personality disorders

Fourteen studies provided data comparing the prevalence of personality disorders (including borderline personality disorder and antisocial personality disorder) between ADHD and non-ADHD individuals [12, 16, 23, 24, 30, 33, 34, 38, 41–44, 47, 53]. In general populations, the prevalence of any personality disorders in the non-ADHD group was estimated at 0% [42] to 3.9% [16] compared to 0.31% [42] to 33.8% [16] in the ADHD group. In clinical populations, the prevalence of any personality disorder in the non-ADHD group was estimated at 6.6% [33] to 34.4% [12] compared to 21.9% [34] to 65.95% [12] in the ADHD group. Two studies of incarcerated populations showed differences in the prevalence of borderline personality disorder and antisocial personality disorder. The prevalence of antisocial personality disorder was higher in the ADHD group in both studies [23, 47], and the prevalence of borderline personality disorder was higher in one study [47]. Detailed information from each study is summarized in Table 4.

## Discussion

In our systematic review, we included 32 studies conducted over 16 years dealing with the prevalence of psychiatric comorbidities between adults with and without ADHD. To our knowledge, this is the first systematic review comparing the prevalence of comprehensive comorbid psychiatric disorders between adults with and without ADHD and including both general and clinical populations. Articles published from 2006 to 2022 were included in this review. This might because the interest in adult ADHD had increased based on the results of longitudinal follow-up studies on children and adolescents with ADHD [54, 55].

The studies included in this review most commonly used ASRS as a diagnostic tool for adult ADHD (12 of 32), followed by CADDID and WURS. As a self-report scale, due to its simplicity and cost-effectiveness, ASRS can be preferred by ADHD investigators. However, the variability in the evaluation tools for adult ADHD is thought to be due to the lack of established diagnostic criteria [56]. Many psychiatric disorders are diagnosed based on observation of an assessor and complaint of a patient. The lack of standardized diagnostic tools can lead to over-diagnosis or under-diagnosis of psychiatric disorders [57, 58] and can cause inconsistency in diagnosis, which complicates comparison studies [59, 60]. Heterogeneity of diagnostic tools observed in this study not only prevented meta-analysis, but also might affect the variability in the reported prevalence of adult ADHD.

In addition, the prevalence of ADHD in adults varied from 1.1% [39] to 8.8% [50] in general population samples and from 5.12% [52] to 35.75% [53] in psychiatric population samples. A previous systematic review of ADHD prevalence in and adult psychiatric population shows a similar range, from 6.9% to 38.75% [61]. In this previous study [61], the authors assumed

**Table 3. Studies comparing the prevalence of substance use disorder and gambling disorder between non-ADHD and ADHD subjects.**

| Author, | Year | Country | N | % of male | Age | Assessment of ADHD | Assessment of comorbid psychiatric disorder | Design | Sample | Prev. of ADHD(%) (non-ADHD/ ADHD) | Findings comparing non-ADHD and ADHD and prevalence of comorbid psychiatric disorders | non-ADHD, n (%) vs ADHD, n (%) |
|---|---|---|---|---|---|---|---|---|---|---|---|---|
| **General sample** | | | | | | | | | | | | |
| Cipollone et al [51] | 2020 | US | 18,913 | 88.3% | Mean age 28.72 in non-ADHD, 28.56 in ADHD | ASRS-S (+) | CIDI & CIDI-SAM | Cross-sectional study (All army study) | General sample–Military sample | 6.6% (17,674/ 1,239) | Previous 30-days SUD diagnosis | 714 (4.04%) vs 211 (17.03%) $\chi^2 = 515.36$ |
| | | | | | | | | | | | Lifetime SUD diagnosis | 2,639 (14.93%) vs 503 (40.60%) $\chi^2 = 780.16$ |
| | | | | | | | | | | | Alcohol use (type 2 —Five or more drinks per day-heavy drinking) | 2,064 (12.04%) vs 305 (25.10%) $\chi^2 = 172.07$ |
| Capusan et al [50] | 2019 | Sweden | 18,167 | 40.08% | 20–45 | DSM-IV criteria | SCID-I | Population-based epidemiological study | General population-Swedish Twin Registry | 8.8% (1,598/ 16,569) | Alcohol abuse | OR = 1.88 [1.44, 2.46] |
| | | | | | | | | | | | Alcohol dependence | OR = 3.58 [2.86, 4.49] |
| | | | | | | | | | | | Stimulants | OR = 2.45 [1.79, 3.35] |
| | | | | | | | | | | | Opiates | OR = 1.97 [1.65, 2.36] |
| | | | | | | | | | | | Cannabis | OR = 2.19 [1.80, 2.68] |
| | | | | | | | | | | | Illicit drug use | OR = 2.27 [1.86, 2.76] |
| | | | | | | | | | | | Poly-substance use | OR = 2.54 [2.00, 3.23] |
| | | | | | | | | | | | Poly-substance use including alcohol | OR = 2.78 [2.21, 3.50] |
| Chen et al [31] | 2018 | Norway | 5,551,807 | 50.81% | 18–64 | ICD-9: 314; ICD-10: F90 diagnosis | ICD | cross-sectional study | General sample | 1.1% (61,129/ 5,490,678) | SUD | PR = 9.74 (9.62–9.86) |

*(Continued)*

Table 3. (Continued)

| Author, | Year | Country | N | % of male | Age | Assessment of ADHD | Assessment of comorbid psychiatric disorder | Design | Sample | Prev. of ADHD(%) (non-ADHD/ ADHD) | Findings comparing non-ADHD and ADHD and prevalence of comorbid psychiatric disorders | non-ADHD, n (%) vs ADHD, n (%) |
|---|---|---|---|---|---|---|---|---|---|---|---|---|
| Hesson and Fowler [35] | 2018 | Canada | 16,957 | NA | 20–64 | Self-report of ADHD (diagnosed by a health professional) | WHO-CIDI modified for the needs of CCHS-MH | Case-control study | General sample -national mental health survey | 2.9% (NA) | 12-month Alcohol dependence | 8 (1.7%) vs 27 (5.6%) $\chi2 = 10.83$ .001 |
| | | | | | | | | | | | Cannabis abuse | 3 (0.6%) vs 13 (2.7%) $\chi2 = 6.376$ .012 |
| | | | | | | | | | | | Cannabis dependence | 3 (0.6%) vs 11 (2.3%) $\chi2 = 4.605$ .032 |
| | | | | | | | | | | | Other drug dependence | 3 (0.6%) vs 17 (3.5%) $\chi2 = 10.01$ .002 |
| Yoshimasu et al [16] | 2016 | US | 5,718 | NA | Mean age ADHD 30.2 (SD 1.9) Non-ADHD controls 30.2 (SD 2.0) | Childhood-identified ADHD with M.I.N.I (+) | M.I.N.I. | Case-control study | General population–Birth cohort sample | NA, (68/335) | Alcohol dependence/ abuse | 51 (15.2%) vs 28 (41.2%) OR adj 3.6 [2.0, 6.7] |
| | | | | | | | | | | | Substance dependence/abuse | 22 (6.6%) vs 18 (26.5%) OR adj 4.4 [2.1, 9.1] |
| Park et al [39] | 2011 | South Korea | 6,081 | ADHD + 59.4% ADHD- 50.5% | 18–59 | ASRS-S v 1.1 (+) | K-CIDI (Korean Ver. Of CIDI) | Epidemiological study | General sample | (69/6,012) | Alcohol abuse/ dependence | 16.6% vs 30.4% OR 1.97 [1.14–3.38] |
| | | | | | | | | | | | Nicotine dependence | 7.7% vs 20.3% OR 2.81 [1.50–5.29] |
| | | | | | | | | | | | Pathological gambling | 0.7% vs 1.4% OR 8.43 [2.63–26.96] |
| Miller et al [38] | 2007 | US | 363 | 51.0% | 18–37 | K-SADS & structured interview | SCID–I, SCID–II | Case-control study | General sample- Recruited ADHD vs control group | NA, (152/ 211) | Any ADHD | SUD $\chi2 = 9.22$ |

(Continued)

**Table 3.** (Continued)

| Author, | Year | Country | N | % of male | Age | Assessment of ADHD | Assessment of comorbid psychiatric disorder | Design | Sample | Prev. of ADHD(%) (non-ADHD/ ADHD) | Findings comparing non-ADHD and ADHD and prevalence of comorbid psychiatric disorders | non-ADHD, n (%) vs ADHD, n (%) |
|---|---|---|---|---|---|---|---|---|---|---|---|---|
| Sobanski et al [6] | 2007 | Germany | 140 | 54.3% | Mean age ADHD+ 36.8 (SD 9.0) ADHD- 39.8 (SD 10.0) | WURS-K & BADDS | SCID-I | Case-control study | General sample -referred ADHD vs control group | NA, (70/70) | Substance related disorders total | 5 (7.1%) vs 21 (30.0%) $\chi^2$ = 12.397 |
| | | | | | | | | | | | Substances total | 2 (2.9%) vs 20 (28.5%) $\chi^2$ = 17.806 |
| | | | | | | | | | | | Substance abuse | 2 (2.9%) vs 12 (17.1%) $\chi^2$ = 8.104 |
| | | | | | | | | | | | Substance dependence | 0 (0%) vs 8 (11.4%) $\chi^2$ = 8.612 |
| Kessler et al [36] | 2006 | US | 3,199 | NA | 18–44 | DIS-IV for childhood pathology & ACDS v1.2 (ADHD-RS) | CIDI | Epidemiologic study | General sample- national survey | 2.6% (NA) | Drug dependence | 0.1% vs 4.4% OR 7.9 [2.3–27.3] |
| | | | | | | | | | | | Any substance disorder | 5.6% vs 15.2% OR 3.0 [1.4–6.5] |
| Secnik et al [42] | 2005 | US | 4,504 | 64.3% | 18≤ | ICD-9 | ICD-9 | Case-control study | General sample- HPM database | NA, (2,252/ 2,252) | Drug or alcohol abuse | 1.87% vs 5.11% |

Clinical sample

| Author, | Year | Country | N | % of male | Age | Assessment of ADHD | Assessment of comorbid psychiatric disorder | Design | Sample | Prev. of ADHD(%) (non-ADHD/ ADHD) | Findings comparing non-ADHD and ADHD and prevalence of comorbid psychiatric disorders | non-ADHD, n (%) vs ADHD, n (%) |
|---|---|---|---|---|---|---|---|---|---|---|---|---|
| Valsecchi et al [52] | 2021 | Italy | 590 | 47.2% | 18–70 | ASRS-S v1.1 (+) and DIVA 2.0 both(+) | M.I.N.I Plus | cross-sectional observational study | Psychiatric outpatients | 5.12% (590/ 44) | Substance abuse, lifetime | 15.1% vs 29.6% $\chi^2$ = 6.34 |
| | | | | | | | | | | | Substance abuse, actual | 6.6% vs 25.0% $\chi^2$ = 19.06 |
| | | | | | | | | | | | Substance use, lifetime | 30.5% vs 54.6% $\chi^2$ = 10.84 .001 |
| | | | | | | | | | | | Substance use, actual | 8.3% vs 29.6% $\chi^2$ = 20.93 .000 |
| Woon and Zakaria [45] | 2019 | Malaysia | 120 | 94.2% | 18–65 | CAADID | M.I.N.I | Cross-sectional study | Psychiatric sample -Forensic ward inpatient | 15.8% (101/ 19) | Alcohol abuse | 2 (2.0%) vs 3 (15.8%) 0.028 |

*(Continued)*

**Table 3.** (Continued)

| Author, | Year | Country | N | % of male | Age | Assessment of ADHD | Assessment of comorbid psychiatric disorder | Design | Sample | Prev. of ADHD(%) (non-ADHD/ ADHD) | Findings comparing non-ADHD and ADHD and prevalence of comorbid psychiatric disorders | non-ADHD, n (%) vs ADHD, n (%) |
|---|---|---|---|---|---|---|---|---|---|---|---|---|
| Roncero et al [41] | 2019 | Spain | 726 | 72.5% | 18≤ | ASRS (14≤) | DSM-IV-TR | Cross-sectional study | Psychiatric patients–treatment seeking AUD patients | 21.1% (573/153) | Cannabis dependence | 18% vs 30.9% $\chi^2 = 12.3$ OR 2.04 [1.36, 3.06] |
| | | | | | | | | | | | Cocaine dependence | 24.6% vs 53.3% $\chi^2 = 46.5$ OR 3.5 [2.41, 5.07] |
| | | | | | | | | | | | Smoking dependence | 72.2% vs 82.9% $\chi^2 = 6.9$ OR 1.86 [1.16, 2.98] |
| Reyes et al [48] | 2019 | US | 472 | 64.6% | 18–80 | PRISM | PRISM | Cross-sectional study | Psychiatric sample–inpatient & outpatient with DSM-IV-TR AUD diagnosis | 6.36% (30/442) | Cannabis abuse, Current | 41 (9.3%) vs 8 (26.7%) |
| | | | | | | | | | | | Amphetamine abuse, current | 17 (3.9%) vs 4 (13.3%) |
| | | | | | | | | | | | Opioid abuse, current | 9 (2.0%) vs 3 (10.0%) |
| Leung and Chan [37] | 2017 | Hong Kong | 254 | 28.7% | 18–64 | ASRS-v1.1 ≥17 & SDS ≥5 (Screening) + DIVA 2.0 (Diagnosis) | DSM-5 | cross-sectional cohort study | Psychiatric sample–clinical outpatients | 19.3% (49/205) | Chronic alcohol use | (2.4% vs 8.2%) |
| | | | | | | | | | | | Problematic gambling | (1% vs 2%) |
| | | | | | | | | | | | Active substance use | (3.9% vs. 8.2%) |
| Retz et al [12] | 2016 | Germany | 163 | 86.5% | Mean age 40.2 (SD 9.4) | DSM-5 & WURS-k ≥ 30 | ICD-10 | Cross-sectional study | Psychiatric sample–GD dx according to ICD-10 | 25.2% (41-current ADHD/122) | Substance use disorders | 4'50 (30.7%) vs 19 (46.3%) χ2 = 6.50 |
| Fatséas et al [33] | 2016 | France | 217 | 66.4% | 18–65 | CAADID | DSM-IV for SUD SCID-II for BPD M.I.N.I for others | Cross-sectional cohort study | Psychiatric sample–addiction outpatient clinic | 23.0% (50/167) | Cannabis dependence | 25.9% vs 58.0% |
| Duran et al [32] | 2014 | Turkey | 246 | NA | 18–60 | WURS score >36 & Turgay's Adult ADD/ADHD Evaluation Scale | SCID-I-CV, SCID-II | Cross-sectional study | Psychiatric sample - outpatient visit patient | 15.9% (39/207) | Other Substance Abuse | 12 (5.8%) vs 7 (18.0%) χ2 = 28.81 |

*(Continued)*

**Table 3.** (Continued)

| Author, | Year | Country | N | % of male | Age | Assessment of ADHD | Assessment of comorbid psychiatric disorder | Design | Sample | Prev. of ADHD(%) (non-ADHD/ADHD) | Findings comparing non-ADHD and ADHD and prevalence of comorbid psychiatric disorders | non-ADHD, n (%) vs ADHD, n (%) |
|---|---|---|---|---|---|---|---|---|---|---|---|---|
| Perugi et al [40] | 2013 | Italy | 96 | 59.4% | 18–65 | ASRS v 1.1 (+), & prior age 7 with ADHD sx | DSM-IV | Cross-sectional observation study | Psychiatric sample- Bipolar I, II disorder diagnosed | 19.8% (19/77) | Alcohol | 7 (9.1%) vs 5 (26.3%) $\chi^2 = 4.1$ |
| | | | | | | | | | | | Substance use disorder | 14 (18.2%) vs 8 (42.1%) $\chi^2 = 7.1$ |
| Incarcerated sample | | | | | | | | | | | | |
| Moore et al [47] | 2016 | Australia | 88 | 76% | 18–72 | ASRS-S (+) & M.I.N.I plus (+) | M.I.N.I plus, PDQ-4, SCID-II | Cross-sectional study | Incarcerated sample | 17.0% (15/73) | Benzodiazepine dependence (lifetime) | 13.7% vs 53.3 OR = 5.30 ([1.30, 21.72]) |
| Konstenius et al [23] | 2015 | Sweden | 96 | 0% | Mean age 39.7 | ASRS-S(+) & CAADID | M.I.N.I | Cross-sectional study | Incarcerated sample- only women | 29% (16/40) | Drug dependence | 58% vs 100% |
| Capuzzi et al [24] | 2022 | Italy | 108 | 100% | 18–65 | WURS-25 & ASRS V1.1 | DSM-5 | Cross-sectional study | Incarcerated sample- only men | 32.4% (35/73) | Cannabis use disorder | 40 (54.8%) vs 25 (71.4%) |
| | | | | | | | | | | | Cocaine use disorder | 47 (64.4%) vs 32 (91.4%) |

OR: Odd Ratio, PR: Prevalence Ratio, NA: Not available (not identified in article)

SUD: Substance Use Disorder, AUD: Alcohol use disorder, GD: Gambling disorder CIDI: Composite International Diagnostic Interview, CIDI-SAM: CIDI-Substance Abuse Module, SCID-I: Structured Clinical Interview for DSM-IV Axis I Disorders, WHO-CIDI: World Health Organization version of the Composite International Diagnostic Interview, M.I.N.I: Mini-International Neuropsychiatric Interview, CAADID: Conner's Adult ADHD Diagnostic Interview for DSM-IV, ASRS-S: Adult Self-Report Scale-Screener, ASRS: Adult Self-Report Scale, BADDS: Brown attention deficit disorder scale, PRISM: Psychiatric research interview for substance and mental disorders SDS: Sheehan Disability Scale, DIVA: Diagnostic Interview for ADHD in Adults, WURS: Wender Utah Rating Scale, WURS-k: German short form of the Wender Utah rating scale, PDQ-4: Personality disorder diagnostic questionnaire for the DSM-IV, SCID-1-CV: Structured Clinical Interview for DSM-IV Axis I Disorders, Clinician Version

**Table 4. Studies comparing the prevalence of personality disorder between non-ADHD and ADHD subjects.**

| Author | Year | Country | N | Male; % | Age | Assessment of ADHD | Assessment of comorbid psychiatric disorder | Design | Sample | Prev. of ADHD(%) (non-ADHD/ ADHD) | Findings comparing non-ADHD and ADHD and prevalence of comorbid psychiatric disorders | non-ADHD, n (%) vs ADHD, n (%) |
|---|---|---|---|---|---|---|---|---|---|---|---|---|
| General sample | | | | | | | | | | | | |
| Solberg et al [43] | 2018 | Norway | 1,701,206 | 51.2% | 18≤ | ADHD medication at adult or ADHD diagnosis registered | ICD-10 | Cross-sectional study | General sample | 2.4% (40,103/ 1,661,103) | Personality disorder | Women 14,079 (1.7%) vs 2,428 (13.6%) Men 8909 (1.1%) vs 2030 (9.1%) |
| Yoshimasu et al [16] | 2016 | US | 5,718 | NA | Mean age ADHD 30.2 (SD 1.9) Non-ADHD controls 30.2 (SD 2.0) | Childhood-identified ADHD with M.I.N.I (+) | M.I.N.I. | Case-control study | General population–Birth cohort sample | NA, (68/ 335) | Antisocial personality disorder | 13 (3.9%) vs 23 (33.8%) OR adj 12.2 [5.3, 27.9] |
| Miller et al [38] | 2007 | US | 363 | 51.0% | 18–37 | K–SADS & structured interview | SCID–I, SCID-II | Case-control study | General sample-Recruited ADHD vs control group | NA, (152/ 211) | ASPD (Any ADHD) | χ2 = 7.32 |
| Secnik et al [42] | 2005 | US | 4504 | 64.3% | 18≤ | ICD-9 | ICD-9 | Case-control study | General sample–HPM database | NA, (2,252/ 2,252) | Antisocial disorder Oppositional disorder | 0% vs 0.31% 0.04% vs 0.53% |
| Clinical sample | | | | | | | | | | | | |
| Sánchez-García et al [53] | 2021, | Puerto-rico, Hungary, Australia | 402 | 79.6% | 18–65 | CAADID | M.I.N.I Plus | Cross-sectional study | Psychiatric inpatients & outpatients with SUD | 35.75% (257/143) | ASPD BPD | 25.41% vs 53.90% OR 3.26 [2.09, 5.08] 20.82% vs 57.45% OR 5.48 [3.40, 8.83] |
| Roncero et al [41] | 2019 | Spain | 726 | 72.5% | 18≤ | ASRS (14≤) | DSM-IV-TR | Cross-sectional study | Psychiatric patients–treatment seeking AUD patients | 21.1% (573/ 153) | Any personality disorder | 14.8% vs 37.4% χ² = 38.17 .0001 OR 3.45 [2.29, 5.17] |
| Gorlin et al [34] | 2016, | US | 1,134 | 42% | Mean age 39.7 (SD 14.4) | DSM-IV based semi-structured clinical interview | SCID | Cross-sectional study | Psychiatric sample-clinical outpatient | 18.0% (204/ 903) | Borderline personality disorder | 7.6% vs 21.9% OR = 3.11 (2.02–4.76) |

*(Continued)*

**Table 4.** (Continued)

| Author | Year | Country | N | Male; % | Age | Assessment of ADHD | Assessment of comorbid psychiatric disorder | Design | Sample | Prev.of ADHD(%) (non-ADHD/ ADHD) | Findings comparing non-ADHD and ADHD and prevalence of comorbid psychiatric disorders | non-ADHD, n (%) vs ADHD, n (%) |
|---|---|---|---|---|---|---|---|---|---|---|---|---|
| Retz et al [12] | 2016 | Germany | 163 | 86.5% | Mean age 40.2 (SD 9.4) | DSM-5 & WURS-k $\geq$ 30 | ICD-10 | Cross-sectional study | Psychiatric sample–GD dx according to ICD-10 | 25.2% (41-current ADHD/ 122) | Personality disorders Cluster B | 56 (34.4%) vs 27 (65.9%) $\chi 2 = 26.84$ 11 (6.7%) vs 7 (17.1%) $\chi 2 = 30.49$ |
| Fatséas et al [33] | 2016 | France | 217 | 66.4% | 18–65 | CAADID | DSM-IV for SUD SCID-II for BPD M.I.N.I for others | Cross-sectional cohort study | Psychiatric sample–addiction outpatient clinic | 23.0% (50/ 167) | Antisocial personality disorder Borderline personality disorder | 6.6% vs 26.0% 13.0% vs 34.7% |
| van Emmerik-van Oortmerssen et al [44] | 2014 | Australia, Belgium, France, Hungary, Netherlands, Norway, Spain, Sweden, Switzerland, US (IASP study) (37) | | | 18–65 | CAADID | M.I.N.I Plus SCID-II | Cross-sectional study | Psychiatric sample-treatment-seeking SUD patients | 13.9% (168/ 1,037) | ASPD BPD | 17.0% vs 51.8% OR 2.8 [1.8–4.2] -alcohol 8.2% vs 34.5% OR 7.0 [3.1–15.6] -drugs 16.7% vs 29.0% OR 3.4 [1.8–6.4] |
| Olsson et al [30] | 2022 | Sweden | 804 | 67.3% | 18≤ | ICD-10 or prescription of ADHD medication | ICD-10 | Cross-sectional study | Psychiatric sample–psychiatric emergency patient with NSSI | 11.6% (711/ 93) | Personality disorder | 142 (20%) vs 28 (30), $\chi 2 = 5.07$ |
| Incarcerated sample | | | | | | | | | | | | |
| Moore et al [47] | 2016 | Australia | 88 | 76% | 18–72 | ASRS-S (+) & MINI plus (+) | M.I.N.I plus, PDQ-4, SCID-II | Cross-sectional study | Incarcerated sample | 17.0% (15/ 73) | BPD ASPD | 13.7% vs 60.0% OR = 7.34 ([1.72, 31.37]) 27.4% vs 93.3% OR = 26.00 ([2.58, 262.30]) |
| Konstenius et al [23] | 2015 | Sweden | 96 | 0% | Mean age 39.7 | ASRS-S (+) & CAADID | M.I.N.I | Cross-sectional study | Incarcerated sample- only women | 29% (16/40) | ASPD | 30% vs 81% |

(*Continued*)

**Table 4.** (Continued)

| Author | Year | Country | N | Male; % | Age | Assessment of ADHD | Assessment of comorbid psychiatric disorder | Design | Sample | Prev.of ADHD(%) (non-ADHD/ ADHD) | Findings comparing non-ADHD and ADHD and prevalence of comorbid psychiatric disorders | non-ADHD, n (%) vs ADHD, n (%) |
|---|---|---|---|---|---|---|---|---|---|---|---|---|
| Capuzzi et al [24] | 2022 | Italy | 108 | 100% | 18–65 | WURS-25 & ASRS V1.1 | DSM-5 | Cross-sectional study | Incarcerated sample- only men | 32.4% (35/ 73) | Personality disorders | 26 (35.6%) vs 21 (60.0%) |

OR: Odd Ratio, NA: Not available (not identified in article)

SUD: Substance Use Disorder, AUD: Alcohol use disorder, GD: Gambling disorder BPD: Borderline Personality Disorder, ASPD: Antisocial personality disorder

SCID: Structured Clinical Interview for DSM-IV, M.I.N.I: Mini-International Neuropsychiatric Interview, ASRS-S: Adult Self-Report Scale-Screener, ASRS: Adult Self-Report Scale, CAADID: Conner's Adult ADHD Diagnostic Interview for DSM-IV, WURS-k: German short form of the Wender Utah rating scale, K-SADS: Kiddie Schedule for Affective Disorders and Schizophrenia, PDQ-4: Personality disorder diagnostic questionnaire for the DSM-IV

HPM: Health and Productivity Management

ISAP: International ADHD in Substance use disorder Prevalence

that this variation might be due to the diversity of diagnostic methods and the inclusion and exclusion criteria in the studies. Similarly, in our study, the aforementioned variability of diagnostic methodologies for ADHD might have affected this various range of prevalence. In a general population, the estimated mean prevalence rate of ADHD in adults was 2.8% in a previous study [15]. Except in two studies included in our review, targeting special populations of army soldiers [51] and twins of Sweden [50], which had higher than estimated prevalence, the range of ADHD prevalence in the general population was 1.1% to 2.9% in our study, similar to that previously observed [11, 56].

The most frequent comorbid psychiatric disorder in the ADHD group was SUD, its prevalence ranging from 2.3% and 41.2% in the general population and between 10.0% and 82.9% in the clinical population; two of three studies showed significant prevalence difference between ADHD and non-ADHD subjects. This finding correlates with a previous meta-analysis that reported that almost one out of every four adolescent and adult patients with SUD presents with ADHD [62, 63], which supports SUD as one of the most frequent comorbid psychiatric conditions in adult ADHD. There are some theoretical opinions of shared key characteristics and pathophysiology between ADHD and SUD, like dopaminergic dysregulation of motivational and reward systems, or reduced frontal function including executive functions and response inhibition [64, 65]. In addition, considering that childhood ADHD is a prominent risk factor for substance misuse and development of SUD due to the most frequent comorbidities in childhood ADHD, like conduct disorder or oppositional defiant disorder [66], untreated and preserved ADHD in adults might have influenced the cross-sectional difference of prevalence rate between ADHD and non- ADHD patients.

Mood disorders, including depressive disorders and bipolar disorders, were also frequently observed comorbid psychiatric disorders in ADHD subjects compared to non-ADHD subjects. The estimated prevalence of depressive disorders in the ADHD group ranged from 8.6% to 55% in the general population and 15.4% to 39.7% in the clinical population. Also, the prevalence of bipolar disorder in the ADHD group was estimated at 4.45% to 35.3% in the general population and 7.4% to 80.8% in the clinical population. For depression, previous studies have also shown a higher prevalence of depressive disorders in young adults with ADHD compared

to non-ADHD subjects as well as higher risk of suicidal behavior [67, 68]. This can be explained by a previous cross-sectional study showing the association between ADHD symptoms and depressive symptoms in young adults as identified by low hedonic responsivity [69]. In addition, according to biologic aspects of depression and ADHD, the two disorders might share similar pathophysiologic regions of the brain including decreased activity in the prefrontal [70, 71], amygdala, and hippocampus regions [72–74]. Furthermore, 10 studies showed a higher prevalence of bipolar disorder, including current hypomania diagnosed by SCID-II, in ADHD subjects than in non-ADHD subjects. Considering that the worldwide prevalence rate of bipolar disorder is estimated as 1–3% [75, 76], which was similar to that of the non-ADHD general population in our study, the prevalence of bipolar disorder in the ADHD group was greater than 3% in all 10 studies. This finding correlated with previous studies reporting reciprocal high comorbidity rates between ADHD and bipolar disorder, which suggests possible shared genetic effects or diagnostic overlap between the two disorders [77].

In anxiety disorder, almost two of three studies showed a higher prevalence in the ADHD group than the non-ADHD group. The prevalence rate in the ADHD group was estimated to range from 4.3% to 47.1% in the general population and from 3.9% to 84% in the clinical population. Only one study of a clinical population dealing with psychiatric outpatients in the US [34] showed a higher prevalence of adjustment disorder in the non-ADHD group. These findings correlate with previous studies that revealed a high prevalence of anxiety in the adult ADHD population [78, 79]. ADHD seems to show different characteristics from anxiety, namely fearlessness and impulsivity. Therefore, various theories have been suggested to explain this phenomenon using developmental or biologic aspects in children and adolescents [80]. Similarly, in adults, as far as we know, the two disorders have been related to several common neuroanatomical regions like the dorsolateral prefrontal cortex or the anterior cingulate cortex, which are critically involved with the executive function control network [10]. In addition, considering a previous study about increased risk of accidents in ADHD over the lifespan [81], traumatic events might have influenced the higher prevalence of anxiety disorders. From a developmental viewpoint, as in depressive disorders, this frequently higher prevalence of anxiety disorder might represent the social and relational difficulties induced by ADHD.

The estimated prevalence of personality disorders in the ADHD group was ranged from 0.31% to 33.8% in the general population and from 21.9% to 65.95% in the clinical population. Previous studies have reported that personality disorders, mostly cluster B or C personality disorders, are present in almost 50% of adults with ADHD [82]. The association between ADHD and personality disorders might be mediated by the symptomatic dimensions of ADHD such as emotional dysregulation and oppositional symptoms [83]. In our review, most studies showed a higher prevalence of cluster B personality disorders in ADHD than in non-ADHD groups. Specifically, in the clinical population, more than 20% of adult ADHD subjects were estimated to have comorbid cluster B personality disorder including borderline personality disorder and antisocial personality disorder. Additionally, most clinical population studies included patients diagnosed with substance use disorders, which correlates with previous observational studies of young male adults with ADHD that revealed associations of antisocial personality disorder with ADHD [84].

## Limitations

There are several limiting factors in this review. As mentioned previously, there is significant heterogeneity across studies diagnosing both ADHD and comorbid psychiatric disorders. This prohibited meta-analysis. Furthermore, except for two international studies [44, 53], the included studies were conducted in high-income regions like Europe or North America. In a

previous epidemiologic study investigating cross-national ADHD prevalence in adults [15], the prevalence differed by country income, with higher rates observed in higher-income countries. This difference in prevalence among countries affects the degree of interest in the disease, which might be why the included studies were mainly conducted in Europe and North America. In addition, we did not differentiate patients according to ADHD or comorbid psychiatric disorder treatment status, which might also have affected the prevalence of comorbid psychiatric disorders. Of the included studies, most explored the prevalence cross-sectionally, so we could not infer a correlation or antecedent relationship between ADHD and comorbidities. Only limited estimates of the associations between ADHD and comorbidities can be provided by our review at the study level. Additionally, we did not assess the risk of bias in each study. To include as many studies as possible to reflect broad and various studies of different countries, populations, and comorbid psychiatric disorders, we omitted the risk of bias assessment. Also, there was no pre-registration for our systematic literature review. These points are limitations to this study.

## Conclusion

In conclusion, our findings indicate a higher prevalence of comorbid psychiatric disorders in ADHD subjects compared to non-ADHD subjects, whether they were previously diagnosed with other psychiatric disorders or not. Furthermore, our results suggest a complex association between the multiple comorbidities of ADHD. Given that ADHD is often unrecognized and under-diagnosed in adults, screening for ADHD might be beneficial for patients presenting multiple psychopathologies including substance abuse, mood disorders, and anxiety disorders. In the future, research on standardization of ADHD diagnosis in adults and its comorbid psychopathologies will be required to distinguish adult ADHD from comorbid psychiatric disorders, and to enable comparison among study conditions. This standardization will aid in differential diagnosis and allow provision of earlier treatment in adult ADHD. In addition, research on the neurobiological and developmental bases of ADHD and its comorbid psychiatric disorders should continue to improve the understanding of the connectivity and associations between various comorbid psychiatric disorders and ADHD in adults.

## Supporting information

**S1 Table. PRISMA checklist.**
(DOCX)

**S1 Text. Article search strategy.**
(DOCX)

## Author Contributions

**Conceptualization:** Young Sup Woo, Won-Myong Bahk.

**Data curation:** Won-Seok Choi, Young Sup Woo.

**Formal analysis:** Won-Seok Choi, Young Sup Woo.

**Supervision:** Won-Myong Bahk.

**Writing – original draft:** Won-Seok Choi, Young Sup Woo.

**Writing – review & editing:** Sheng-Min Wang, Hyun Kook Lim.

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
