## [Decision Letter · Decision Letter 0]

25 Jul 2022

PONE-D-22-17093The Prevalence of Psychiatric Comorbidities in Adult ADHD Compared With Non-ADHD Populations : A Systematic Literature ReviewPLOS ONE

Dear Dr. Bahk,

Thank you for submitting your manuscript to PLOS ONE. After careful consideration, we feel that it has merit but does not fully meet PLOS ONE’s publication criteria as it currently stands. Therefore, we invite you to submit a revised version of the manuscript that addresses the points raised during the review process.

 Dear Authors Thanks for submitting this paper to our journal The paper has significant value in adding to existing science however it needs modification as suggested by our reviewers There are 4 reviewers as its a systematic reviewI would like you to modify the paper and incorporate all the suggestions if possibleregards and good luck 

We look forward to receiving your revised manuscript.

Kind regards,

Soumitra Das

Academic Editor

PLOS ONE

Journal Requirements:

2. We note that this manuscript is a systematic review or meta-analysis; our author guidelines therefore require that you use PRISMA guidance to help improve reporting quality of this type of study. Please upload copies of the completed PRISMA checklist as Supporting Information with a file name “PRISMA checklist”.

Reviewers' comments:

Reviewer's Responses to Questions

**Comments to the Author**

1. Is the manuscript technically sound, and do the data support the conclusions?

Reviewer #1: Partly

Reviewer #2: Yes

Reviewer #3: Partly

Reviewer #4: Yes

2. Has the statistical analysis been performed appropriately and rigorously? 

Reviewer #1: N/A

Reviewer #2: Yes

Reviewer #3: N/A

Reviewer #4: Yes

3. Have the authors made all data underlying the findings in their manuscript fully available?

Reviewer #1: Yes

Reviewer #2: Yes

Reviewer #3: Yes

Reviewer #4: No

4. Is the manuscript presented in an intelligible fashion and written in standard English?

Reviewer #1: Yes

Reviewer #2: Yes

Reviewer #3: Yes

Reviewer #4: Yes

5. Review Comments to the Author

Reviewer #1: Thank you for the opportunity to review, 'The Prevalence of Psychiatric Comorbidities in Adult ADHD Compared With Non-ADHD Populations : A Systematic Literature Review'. The manuscript describes a systematic review of the relation between psychiatric comorbidities and Adult ADHD. The authors found the most frequent comorbid psychiatric disorder in the ADHD group was substance use disorder (SUD), followed by mood disorders, anxiety disorders, and personality disorders. This review covers an important area of research in need of synthesis and guidance for future studies. This review also has a particular strength in synthesizing data into four psychiatric categories (4 tables) and across three population groups. However, there are several significant concerns that affect the review's contribution in its current state. One overall major concern is many important components of a systematic review process were either not reported, poorly structured, or presented, in accordance with PRISMA protocol. There are insufficient details of the methods and analysis provided to allow replication. It also makes the establishment of arguments and synthesis of data very difficult to comprehend. The most notable concerns are described below:

1. P. 10, More information is needed to describe the study selection process, e.g. The number of researchers who conducted the screening of titles and abstracts, and the process of managing disagreement between the reviewers during the screening process. The range of publication years that were searched.

2. P. 11, '.....and the full-text articles were reviewed randomly by two researchers....' needs further clarification. It is unclear what 'randomly' means. Did both researchers review all the full-text articles? The percentage of full-text articles reviewed by each reviewer should be reported. The percentage of inclusion consistency between the two reviewers should also be mentioned.

3. P. 12, whereas a clinical group study was defined, it is suggested that a general population group study should also be defined. '...A clinical group study was defined as one in which the study population included patients who had previously been clinically diagnosed with any psychiatric disorder....' This study examined 'psychiatric comorbidities', readers would expect that a certain percentage of individuals in the 'general population' may also have a psychiatric disorder. Thus, a clear explanation to distinguish both may be needed. The authors also mentioned 'incarcerated people', a clear definition of incarcerated population specific to the current study should also be considered. In the results section, the authors actually classified studies into three categories, i.e. Clinical, general, and incarcerated. In this case, further descriptions of the incarcerated population would be required in both the introduction and method sections.

4. Given the search was completed about one year ago (22 July, 2021) (p.10), the authors may consider updating the search.

5. According to PRISMA checklist, a risk of bias assessment should be conducted and reported in both Methods and Results sections; which is not presented in the current manuscript.

6. Suggest the authors report if the review was pre-registered in the method section.

7. Suggest updating the PRISMA 2020 checklist (S1 Table): e.g. '20a Results of syntheses' were presented on p.13-15, now is presented as NA

8. There is room to improve the presentation of the tables to increase clarity, e.g. The information presented in 'Findings comparing non-ADHD and ADHD and prevalence of comorbid psychiatric disorders non-ADHD, n (%) vs ADHD, n (%)' could be presented in separate columns; consider to delete the comment column and to use a footnote to provide an additional comment if needed; author year and country could be reported in separate columns, consider not to report the number of male participants but has a separate column to report the percentage of male, etc.

9. In the discussion section, suggest the findings of different use of diagnostic tools, i.e. '...in this review......in adults' p.37, should be reported in the results section. Those information was not mentioned in the results section previously. The authors mentioned in the abstract that 'Standardization of ADHD diagnostic tools is necessary in the future'(p.8). Arguments to support this conclusion needs to be further established in the discussion and conclusion sections.

10. In the limitations section, the authors discussed heterogeneity across studies and other risks of bias issues (P. 41). However those concepts were not thoroughly explored before. Suggest the authors consider reporting that information in the results section, under risk of bias in studies and/or reporting biases. In the limitations section, the authors may consider suggesting future research directions to address the current limitations.

11. In the conclusion section (p.35-36), the authors provided suggestions of future research (.e.g.... to clarify the aspects to treat and improve the quality of life and the functional and psychosocial impairments of individuals with ADHD....), however, those recommendations seem not to be strongly linked to/supported by the current findings. The authors may need to consider establishing a strong conclusion based on the current key findings and the original research question.

Reviewer #2: 1. This systematic review is based on heterogenous sample of studies which results in significant heterogeneity in the results obtained but this has been identified in the limitations section.

2. In my opinion the statistical analysis performed is standard and appropriate

3. The authors have provided access to all their findings.

4. The manuscript has been presented well in with a good written standard of English.

Reviewer #3: Thank you for the opportunity to revise the systematic review manuscript. I thoroughly enjoyed it. Overall, the manuscript has been well written. However, I believe the authors could further improvise the manuscript if they consider responding to the following comments.

Major comments

1. Introduction:

The Introduction section mentions that the aim of the review was to estimate the prevalence of comorbid psychiatric disorders in adults with and without ADHD. However, the authors use varying terms in the texts such as degree, incident rates and prevalence rates at different instances. I suggest the authors clarify this. Also, please strengthen the rationale for the need of this systematic review.

2. Methods:

The authors mention following Preferred Reporting Items for Systematic Reviews and Meta-analysis (PRISMA) as the methodology and the flow diagram is also provided. That is good.

My concern is if the authors could provide detailed information on the PRISMA guide in terms of the population characteristics, comparators, outcome, studies etc. I suggest the authors to include this information in the methods section.

The authors quoted five electronic libraries to be used while searching for the relevant articles. However, the search term given is Prevalence AND (ADHD OR ADD OR Attention Deficit) AND Adult AND (comorbidity or comorbid) in titles or abstracts. To my knowledge, the search terms operate differently for different databases, so please clarify on the search terms used to screen for articles in the 5 databases.

Study selection:

While the authors mention the initial inclusion/ exclusion review was conducted, I suggest the authors give information on the specific inclusion and exclusion criteria. Also, the authors mention relevance of the article and describe that if they were found irrelevant, full texts were reviewed. Please provide what do the authors mean when they say relevant articles in the context of the review.

Did the authors include all study types, except for systematic reviews and meta-analyses? I would suggest the authors to be explicit in which study types were included and if they were determined before the study started.

Data collection:

The full text articles were reviewed randomly by two researchers (WSC, YSW). Please clarify what the authors meant by saying randomly. Was it not decided beforehand who will do what in the review process amongst the authors?

3. Results

Tables:

The authors have included several studies where age groups are < 18 years while, they mention one of the inclusion criteria was age more than 18 years.

Also, for other studies included there were those aged 16-65 and 16-70 years. Please specify the age groups for inclusion since the study focus was on adults.

4. Discussion

The authors are advised to discuss on why the studies of 15 years are included. In other words, what was the earliest year of study inclusion? If you did not find studies before that year, what could be the reasons?

The authors are also suggested to cite the articles used in the text. For instance, when the authors say that the range of ADHD prevalence in the general population was 1.1-2.9% in their study, similar to previously observed.

The authors cite social and relational difficulties in ADHD as possible reasons for increased frequency of anxiety disorders. Please clarify on this statement.

I also suggest the authors to discuss on what could be the reasons for studies mostly available from high income countries such as Europe and North America.

5. Other important points

Systematic review protocols are required to be registered before you start the review process to avoid duplications and assist in transparency of the work. There are various options for registration such as PROSPERO, the Registry of Systematic Reviews/Meta-Analyses in Research Registry etc.

In the manuscript submitted, I do not see the registration number. Could the authors include the registration number and the registry where the protocol has been registered.

The authors have not described if they have used quality control for the studies included. In other words, risk of bias assessment is one of the most important steps in any systematic review and metanalysis. Also, who did the quality assessment of the studies included?

Minor comments

The article type mentioned in the manuscript file is research article, while the manuscript submitted is systematic review. So, I suggest the authors to change the name of the article type.

In the title, the authors mention systematic literature review instead if just systematic review. Do the authors have specific reason for this?

There are several grammatical errors throughout the text. I suggest the authors do a thorough grammatical check and proof reading while reviewing the manuscript.

Reviewer #4: This study examines the prevalence of comorbid psychiatric disorders in ADHD populations and contributes to the evidence base in this area through the systematic review conducted by the authors of the paper.

The introduction is adequate - but there are minor errors that should be corrected i.e. replace impulsiveness with impulsivity, add 'with' following presenting, replace high incident rates with 'high incidence rates' to correct these errors.

Methodology - adequately explained - but again, there are errors in the first sentence 'the methodology of the present review was followed by PRISMA' should be replaced by 'the methodology of the present review followed PRISMA protocol'. I would recommend here that the statistical analyses used should be explained further.

Results - I would recommend adding a section on psychotic illnesses and personality disorders as these are relevant and included in the results table.

Discussion - This section is adequate, but again has a few errors - replace 'variety' by 'variation', rewrite the sentence on 'the most frequent comorbid psychiatric disorder is SUD, ranging from...' include 'its prevalence ranging from...'

The article would benefit from proofreading to correct grammatical and syntactical errors.

A major limitation is the heterogeneity of these studies - this has been acknowledged.

Overall, I think the article addresses an important area and I recommend revising it for publication.

6. PLOS authors have the option to publish the peer review history of their article (what does this mean?). If published, this will include your full peer review and any attached files.

Reviewer #1: No

Reviewer #2: No

Reviewer #3: **Yes: **Saraswati Dhungana

Reviewer #4: **Yes: **Karishma Kulkarni

---

## [Author Response · Author response to Decision Letter 0]

28 Aug 2022

Responses to the Reviewer’s comment

28 August, 2022

Dear reviewers and editorial staffs of PLOS ONE

We are sincerely grateful for your thorough consideration and scrutiny of our manuscript, “The Prevalence of Psychiatric Comorbidities in Adult ADHD Compared With Non-ADHD Populations : A Systematic Literature Review”, control number PONE-D-22-17093. Through the accurate comments made by the reviewers, we better understand the critical issues in this paper. We have revised the manuscript according to the Reviewer’s suggestions. We hope that our revised manuscript will be considered and accepted for publication in the PLOS ONE. We acknowledge that the scientific and clinical quality of our manuscript was improved by the scrutinizing efforts of the reviewers and editors.

The changes within the revised manuscript were highlighted in blue at 'Response to Reviewers. docx' file. Point-by-point responses to the reviewers’ comments are provided below.

Reviewer #1:

1. P. 10, More information is needed to describe the study selection process, e.g. The number of researchers who conducted the screening of titles and abstracts, and the process of managing disagreement between the reviewers during the screening process. The range of publication years that were searched.

Author’s response : We appreciate the reviewer’s comment. We have now presented the number and rolls of researchers participated in study selection and data extraction process. We also added the range of publication “from 1 January 1990 to 22 July 2021”. “prior to June 22, 2021” which represented the search period before the revision process, thought to be an ambiguous description. Thank you for the reviewer’s comment for accurate description of the searching period.

2. P. 11, '.....and the full-text articles were reviewed randomly by two researchers....' needs further clarification. It is unclear what 'randomly' means. Did both researchers review all the full-text articles? The percentage of full-text articles reviewed by each reviewer should be reported. The percentage of inclusion consistency between the two reviewers should also be mentioned

Author’s response : We appreciated the reviewer’s comment. We wanted to mean that both two researchers reviewed all article which was selected by them in any order. Therefore, we changed these description more exactly and clarify the sentence. We also specified the percentage of inclusion consistency reviewed by each investigator at ‘Study selection’ section. Thank you for the reviewer’s consideration for accurate description of study selection process. The revised study selection and data collection process section is shown below :

…Microsoft Excel was used to develop a data extraction sheet, and the whole included full-text articles were reviewed randomly by two researchers (WSC, YSW) in duplicate who also conducted the initial data search and study selection process.

3. P. 12, whereas a clinical group study was defined, it is suggested that a general population group study should also be defined. '...A clinical group study was defined as one in which the study population included patients who had previously been clinically diagnosed with any psychiatric disorder....' This study examined 'psychiatric comorbidities', readers would expect that a certain percentage of individuals in the 'general population' may also have a psychiatric disorder. Thus, a clear explanation to distinguish both may be needed. The authors also mentioned 'incarcerated people', a clear definition of incarcerated population specific to the current study should also be considered. In the results section, the authors actually classified studies into three categories, i.e. Clinical, general, and incarcerated. In this case, further descriptions of the incarcerated population would be required in both the introduction and method sections.

Author’s response : We appreciate for your suggestion. We have clarified how to define study groups. The core criterion for classifying study group is whether any psychiatric disorders except ADHD were diagnosed for whole study population before the each study started. That is why we descripted the prevalence of psychiatric disorders except ADHD separately at each population groups. And also we have added definition for ‘incarcerated population’ method section. Thank you for the reviewer’s comment for accurate description of the classification of studies. The revised text for description of target population is shown below :

…A general group study was defined as a study in which the whole study population was not diagnosed with any psychiatric disorders before the start of each study.

…The incarcerated study group was defined as the study only included those incarcerated in prison facility as participants.

4. Given the search was completed about one year ago (22 July, 2021) (p.10), the authors may consider updating the search.

Author’s response : We appreciate for your suggestion. We updated the searched articles after 22 July 2021. We conducted searches up to August 1, 2022 using the same inclusion & exclusion criteria for each database. As a result, 2 more studies were included to our review. We also updated the flow chart, tables and manuscript. Updated numbers and searching strategy of articles are presented at Supplement 2.

5. According to PRISMA checklist, a risk of bias assessment should be conducted and reported in both Methods and Results sections; which is not presented in the current manuscript.

Author’s response : We appreciate for your comment. We agree that assessment of ROB (risk of bias) in each studies is important and should be conducted for high quality systematic reviews. However, in our review, we did not performed meta-analysis or statistical synthesis with extracted data, but we summarized and re-arranged the prevalence of comorbid psychiatric disorders in multiple studies comparing ADHD and non-ADHD adults to examine the trend of the difference of each comorbid psychiatric disorders. In order to include as many studies as possible to reflect broad and various studies of different countries, populations and comorbid psychiatric disorders, we did not assess the quality or ROB of each included studies. This is also why we presented our study as a systematic literature review. Consequently, this is critical limitation point of our review. Therefore, we revised our limitation section that we did not evaluated the risk of bias. Instead, we assessed the risk of bias of our review using AMSTAR-2 tool, and presented it in limitation section. We hope that this could be the explanation for not assessing ROB of each studies which could affect the limited results of our review. The revised text for limitation section is shown below :

…Additionally, we did not assess the risk of bias in each studies. In order to include as many studies as possible to reflect broad and various studies of different countries, populations and comorbid psychiatric disorders, we skipped the risk of bias assessment. When our review was evaluated by AMSTAR-2 (85), which evaluates the quality of systematic review studies, our review was rated as low quality review. 

85. Shea BJ, Reeves BC, Wells G, Thuku M, Hamel C, Moran J, et al. AMSTAR 2: a critical appraisal tool for systematic reviews that include randomised or non-randomised studies of healthcare interventions, or both. BMJ. 2017;358:j4008. doi: 10.1136/bmj.j4008.

6. Suggest the authors report if the review was pre-registered in the method section.

Author’s response : We appreciate for your comment. We agree that pre-registration needed to improve the quality and reliability of research. However, due to our study was systematic literature review scoping for summarizing and providing overview about prevalence of comorbid psychiatric disorders in adult ADHD, we could not register our study in PROSPERO. This point is important limitation of this study, so we presented this points in limitation section. The revised text for limitation section is shown below. 

…Also, there was no pre-registration for our systematic literature review. These points raise limitations to this study.

7. Suggest updating the PRISMA 2020 checklist (S1 Table): e.g. '20a Results of syntheses' were presented on p.13-15, now is presented as NA

Author’s response : We appreciate for your comment. We have confirmed the error in PRISMA 2020 checklist and updated that part as NA to p.13-15. Thank you for the opportunity to correct the mistake according to the reviewer's point.

8. There is room to improve the presentation of the tables to increase clarity, e.g. The information presented in 'Findings comparing non-ADHD and ADHD and prevalence of comorbid psychiatric disorders non-ADHD, n (%) vs ADHD, n (%)' could be presented in separate columns; consider to delete the comment column and to use a footnote to provide an additional comment if needed; author year and country could be reported in separate columns, consider not to report the number of male participants but has a separate column to report the percentage of male, etc.

Author’s response : We appreciate for your suggestion. We agree with reviewer and edited our tables as recommended to improve the presentation of results clearly. We divided author, year and country of study in different column. And also we separated 'Findings comparing non-ADHD and ADHD and prevalence of comorbid psychiatric disorders non-ADHD, n (%) vs ADHD, n (%)' section as two divided columns for clarifying the results. And we deleted the comment column and switch the information to footnotes. Thank you for the reviewer’s comment for clarifying the table of study results.

9. In the discussion section, suggest the findings of different use of diagnostic tools, i.e. '...in this review......in adults' p.37, should be reported in the results section. Those information was not mentioned in the results section previously. The authors mentioned in the abstract that 'Standardization of ADHD diagnostic tools is necessary in the future'(p.8). Arguments to support this conclusion needs to be further established in the discussion and conclusion sections.

Author’s response : We appreciate for your suggestion. We modified our manuscript as the reviewer suggested. In result section, we added ‘Diagnostic tools of included studies’ paragraph separately and presented the results. And in discussion section, we established some arguments to support our conclusion. According to previous studies, heterogeneity of diagnostic tools can cause inconsistencies in diagnosis. This also makes it difficult to compare the diagnostic characteristic of study population across studies. We added the reference for the need of standardized diagnosis in adult ADHD for suitable comparison of studies about disease prevalence. The revised discussion section is shown below : 

However, variability in the evaluation tools for adult ADHD was observed and is thought to be due to the lack of established diagnostic criteria for ADHD in adults, which was previously argued in systematic review about ADHD prevalence in adults (56). Psychiatric disorder is characterized by being diagnosed with observation of assessor and complaint of patient. Non-standardized diagnostic tools can lead to over-diagnosis or under-diagnosis of psychiatric disorders (57, 58) , and also cause inconsistency in diagnosis which makes comparisons between studies difficult. (59, 60) Heterogeneity of diagnostic tools observed in this study not only made meta-analysis impossible, but also might affect the diversity of the prevalence of adult ADHD.

 56. Song P, Zha M, Yang Q, Zhang Y, Li X, Rudan I. The prevalence of adult attention-deficit hyperactivity disorder: A global systematic review and meta-analysis. J. Glob. Health. 2021;11.

57. Zimmerman M. A Review of 20 Years of Research on Overdiagnosis and Underdiagnosis in the Rhode Island Methods to Improve Diagnostic Assessment and Services (MIDAS) Project. Can J Psychiatry. 2016;61(2):71-9. doi: 10.1177/0706743715625935. PubMed PMID: 27253697; PubMed Central PMCID: PMCPMC4784239.

58. Newson JJ, Hunter D, Thiagarajan TC. The heterogeneity of mental health assessment. Front. Psychiatry. 2020;11:76.

59. Wisco BE, Miller MW, Wolf EJ, Kilpatrick D, Resnick HS, Badour CL, et al. The impact of proposed changes to ICD-11 on estimates of PTSD prevalence and comorbidity. Psychiatry Res. 2016;240:226-33.

60. Linden M, Muschalla B. Standardized diagnostic interviews, criteria, and algorithms for mental disorders: garbage in, garbage out. Eur Arch Psychiatry Clin Neurosci. 2012;262(6):535-44.

10. In the limitations section, the authors discussed heterogeneity across studies and other risks of bias issues (P. 41). However those concepts were not thoroughly explored before. Suggest the authors consider reporting that information in the results section, under risk of bias in studies and/or reporting biases. In the limitations section, the authors may consider suggesting future research directions to address the current limitations.

Author’s response : We appreciate for your suggestion. The bias issue that we mentioned in the limitation section is a wrong presentation of our study. As the reviewer mentioned, we did not assess bias of each included studies which could affect the quality of our systematic literature review. This is one of the important limitations of our study. We now have deleted the sentence at limitation section. 

11. In the conclusion section (p.35-36), the authors provided suggestions of future research (.e.g.... to clarify the aspects to treat and improve the quality of life and the functional and psychosocial impairments of individuals with ADHD....), however, those recommendations seem not to be strongly linked to/supported by the current findings. The authors may need to consider establishing a strong conclusion based on the current key findings and the original research question.

Author’s response : We appreciate for your comment. The sentence was described differently from the meaning we originally intend to suggest. We revised and clarified our suggestion to future research. The revised conclusion section is shown below :

…In the future, research on standardization of ADHD diagnosis in adults and its comorbid psychopathologies is required to distinguish adult ADHD from comorbid psychiatric disorders and to enable comparison between the results from different study conditions. Which could help for differential diagnosis and providing earlier treatment to adult ADHD individual.

Reviewer #2: 

1. This systematic review is based on heterogenous sample of studies which results in significant heterogeneity in the results obtained but this has been identified in the limitations section.

2. In my opinion the statistical analysis performed is standard and appropriate

3. The authors have provided access to all their findings.

4. The manuscript has been presented well in with a good written standard of English.

Author’s response : We appreciate the reviewer’s comment. Thank you for the reviewer's compliment and encouragement of our study. 

Reviewer #3:

1. Introduction:

The Introduction section mentions that the aim of the review was to estimate the prevalence of comorbid psychiatric disorders in adults with and without ADHD. However, the authors use varying terms in the texts such as degree, incident rates and prevalence rates at different instances. I suggest the authors clarify this. Also, please strengthen the rationale for the need of this systematic review.

Author’s response : We appreciate for your comment. We have clarified and replaced the terms that meant the prevalence of psychiatric disorders. The psychiatric disorders associated with adult ADHD have a long-lasting effect on a patient's life, but they are difficult to differentiate with ADHD and presented as turbid and heterogeneous phenotype. Therefore, we clarify and strengthen the rationale of purpose for our study in Introduction section. Revised introduction section is shown below : 

However, to the best of our knowledge, despite the high prevalence of psychiatric disorders documented in previous studies in the adult ADHD subjects (10, 11, 15, 16) and their importance in the clinical field, no systematic literature review has specifically compared the prevalence of comorbid psychiatric disorders between adults with and without ADHD. Considering the high prevalence and the impact on quality of life, establishing a perspective on frequent comorbid psychiatric disorders in adult ADHD would be helpful for individuals with ADHD and clinicians. Thus, the aim of our study was to ascertain the difference in the prevalence rates of comorbid psychiatric disorders between adults with and without ADHD including both clinical and general populations.

2. Methods:

The authors mention following Preferred Reporting Items for Systematic Reviews and Meta-analysis (PRISMA) as the methodology and the flow diagram is also provided. That is good.

My concern is if the authors could provide detailed information on the PRISMA guide in terms of the population characteristics, comparators, outcome, studies etc. I suggest the authors to include this information in the methods section.

Author’s response : We appreciate for your suggestion. We added new information about research question using PICO (Population, Intervention, Comparison, Outcome) of our study and the several components of PRISMA guideline which need to be explained in our study in ‘Study search and data source’ in method section. The revised paragraph is shown below :

…Following PRISMA guideline, we have conducted research question by PICO (Population, Intervention, Comparison and Outcome). Target population were adults with ADHD (P) who were diagnosed with any diagnostic tools (I). We compared the prevalence of comorbid psychiatric disorders between ADHD and non-ADHD patients (C,O).

The authors quoted five electronic libraries to be used while searching for the relevant articles. However, the search term given is Prevalence AND (ADHD OR ADD OR Attention Deficit) AND Adult AND (comorbidity or comorbid) in titles or abstracts. To my knowledge, the search terms operate differently for different databases, so please clarify on the search terms used to screen for articles in the 5 databases.

Author’s response : We appreciate for your suggestion. We agree that additional information for search terms should be provided. We added additional supplement (Supplement 2) that describe the search strategy and numbers of articles for each five database. 

Study selection:

While the authors mention the initial inclusion/ exclusion review was conducted, I suggest the authors give information on the specific inclusion and exclusion criteria. Also, the authors mention relevance of the article and describe that if they were found irrelevant, full texts were reviewed. Please provide what do the authors mean when they say relevant articles in the context of the review.

Author’s response : We appreciate for your suggestion. We agree that additional information for study selection process. We have reviewed abstracts and titles of the selected articles initially. If the information in titles or abstracts of each article were unclear or insufficient, a full-text review for each article was performed to define whether sufficient information were present in main-text. We have clarified the study selection process. The revised study selection section is shown below :

…The initial inclusion/exclusion review was completed based on titles and abstracts, and if the relevance of the article was unclear, a full-text review was performed to define eligibility of each studies.

Did the authors include all study types, except for systematic reviews and meta-analyses? I would suggest the authors to be explicit in which study types were included and if they were determined before the study started.

Author’s response : We appreciate for your suggestion. We agree that additional information for included study types in our review. We clarified the included study types in study selection section. We purposed to include all study types except reviews or meta-analysis because we tried to collect studies which directly comparing the population groups whether diagnosed ADHD or not. As a result, RCTs were not included in our review, and there were only cross-sectional studies. Revised text in study selection section is shown below : 

We selected all types of research including primary studies, but except a systematic review or meta-analysis defined by title and method.

Data collection:

The full text articles were reviewed randomly by two researchers (WSC, YSW). Please clarify what the authors meant by saying randomly. Was it not decided beforehand who will do what in the review process amongst the authors?

Author’s response : We appreciate for your comment. The selected articles were reviewed by both two authors. We have clarified the data collection process and revised the sentence. Revised data collection section is presented at answer for Reviewer #1, Question 2.

3. Results

Tables:

The authors have included several studies where age groups are < 18 years while, they mention one of the inclusion criteria was age more than 18 years.

Also, for other studies included there were those aged 16-65 and 16-70 years. Please specify the age groups for inclusion since the study focus was on adults.

Author’s response : We appreciate for your comment. We agree with that the target population should be clarified in systematic review articles. Our review included studies using adult samples aged 18 or older. In table 1~4, the description “18≤” indicates the age of 18 or older. As a result of reconfirming the included studies we presented in the table 1~4, we could not find any studies involving individuals ranged over 16 years age or older. Please let us know if there is something we haven't checked. Thank you for the opportunity to reconfirm the included studies according to the reviewer’s point.

4. Discussion

The authors are advised to discuss on why the studies of 15 years are included. In other words, what was the earliest year of study inclusion? If you did not find studies before that year, what could be the reasons?

Author’s response : We appreciate for your thoughtful advice. Our publication searching period was prior to 22 July 2021, and we updated to July 2022 according to Reviewer #1’s comment. We tried not to set the initial search period in order to include as many studies as possible dealing with adult ADHD. As a result, the earliest study we included was Secnick et al, (2006). We think that this is because after the publication of follow-up study for children or adolescent ADHD like Biederman, mick & Faraone (2000) et al, the interest on adults with ADHD has increased after that period. We added our opinion in the discussion section.

The authors are also suggested to cite the articles used in the text. For instance, when the authors say that the range of ADHD prevalence in the general population was 1.1-2.9% in their study, similar to previously observed. 

Author’s response : We appreciate the reviewer’s comment. We cited the articles that we referenced (The prevalence of adult attention-deficit hyperactivity disorder: A global systematic review and meta-analysis )in discussion section. 

The authors cite social and relational difficulties in ADHD as possible reasons for increased frequency of anxiety disorders. Please clarify on this statement.

Author’s response : We appreciate the reviewer’s comment. As we reconfirmed the mentioned part, we consider that our statement is lacking of sufficient evidence. Therefore, we removed the sentence “From a developmental…relational difficulties induced by ADHD” and hope that the deletion clarifies the points we attempted to make. 

I also suggest the authors to discuss on what could be the reasons for studies mostly available from high income countries such as Europe and North America.

Author’s response : We appreciate the reviewer’s suggestion. We consider that this was influenced by the cultural/public health environmental difference which might influenced the diagnosis and prevalence of ADHD in adults like in adolescents (Skounti et al,2007). Because of the variety of clinical and pathological features of adult ADHD, the diagnosis of adult ADHD largely dependent on clinical presentation. And also ADHD prevalence may be culture dependent, as what is considering abnormal in one culture may be more acceptable in another, and culture may affect the report by informants or clinicians. This makes the prevalence of adult ADHD differs by countries (Song et al, 2021). In particular, in previous studies related to prevalence of adult ADHD (Fayyad et al, 2017), the prevalence was higher in high-income countries than in low-income countries, which may have caused differences in interest in adult ADHD. This difference of interest about adult ADHD might also influenced the interest in research on comorbidities. We have revised the text and cited references to be more in line with your comments at limitation section additionally. We hope that the revision clarifies the points we attempt to make. Revised limitation section is shown below : 

Considering a previous epidemiologic study investigating cross-national ADHD prevalence in adults (15), the prevalence differed by country income, which was higher in high-income countries, so our findings might be limited. This difference in prevalence among countries affects the degree of interest in the disease, which might be the reason why the included studies were mainly conducted in Europe and North America.

Reference 

Skounti M, Philalithis A, Galanakis E. Variations in prevalence of attention deficit hyperactivity disorder worldwide. Eur J Pediatr. 2007 Feb;166(2):117-23. doi: 10.1007/s00431-006-0299-5. Epub 2006 Oct 11. PMID: 17033803.

Song P, Zha M, Yang Q, Zhang Y, Li X, Rudan I. The prevalence of adult attention-deficit hyperactivity disorder: A global systematic review and meta-analysis. Journal of global health. 2021;11

Fayyad J, Sampson NA, Hwang I, Adamowski T, Aguilar-Gaxiola S, Al-Hamzawi A, Andrade LH, Borges G, de Girolamo G, Florescu S, Gureje O. The descriptive epidemiology of DSM-IV adult ADHD in the world health organization world mental health surveys. ADHD Attention Deficit and Hyperactivity Disorders. 2017 Mar;9(1):47-65.

Minor comments

The article type mentioned in the manuscript file is research article, while the manuscript submitted is systematic review. So, I suggest the authors to change the name of the article type.

In the title, the authors mention systematic literature review instead if just systematic review. Do the authors have specific reason for this?

There are several grammatical errors throughout the text. I suggest the authors do a thorough grammatical check and proof reading while reviewing the manuscript.

Author’s response : We appreciate the reviewer’s comment. As following PLOS ONE’s submission guideline for systematic review, we submitted our article as “Research Article” section. We specified our study as systematic literature review in title because it is a systematic review that summarize the results of previous literatures without performing meta-analysis or other statistical synthesis.

Reviewer #4

The introduction is adequate - but there are minor errors that should be corrected i.e. replace impulsiveness with impulsivity, add 'with' following presenting, replace high incident rates with 'high incidence rates' to correct these errors.

Author’s response : We appreciate the reviewer’s comment. We revised the text that were not grammatically correct. We hope the revised text could clarify the meaning we attempt to make. 

Methodology - adequately explained - but again, there are errors in the first sentence 'the methodology of the present review was followed by PRISMA' should be replaced by 'the methodology of the present review followed PRISMA protocol'. I would recommend here that the statistical analyses used should be explained further.

Author’s response : We appreciate the reviewer’s comment. We revised the text that were not grammatically correct. We agree that the method of statistical analysis should be mentioned in systematic review study. However, we did not performed quantitative synthesis or meta-analysis in our study. For this reason, we did not mentioned specific statistical analysed used in our study. We hope this could be adequate answer for your opinion.

Results - I would recommend adding a section on psychotic illnesses and personality disorders as these are relevant and included in the results table.

Author’s response : We appreciate for reviewer’s suggestion. At first, in Table 4 and ‘Prevalence of personality disorders’ paragraph in Results section, we presented the difference of prevalence between ADHD and non-ADHD adults including antisocial personality disorder and borderline personality disorder. In case of psychotic disorder, there was only one included study reporting the difference of prevalence between ADHD and non-ADHD in adults. For this reason, we believe that there were no sufficient meaning for comparing the prevalence of psychotic disorders between ADHD and non-ADHD, so it was not presented in the manuscript. We hope that this could explain the reason for excepting psychotic disorder in our point of view.

Discussion - This section is adequate, but again has a few errors - replace 'variety' by 'variation', rewrite the sentence on 'the most frequent comorbid psychiatric disorder is SUD, ranging from...' include 'its prevalence ranging from...'

The article would benefit from proofreading to correct grammatical and syntactical errors.

Author’s response : We appreciate the reviewer’s comment. We revised the text that were not grammatically correct. We rechecked our manuscript and corrected few grammatical and syntactical errors. Thank you for the opportunity to correct the mistake according to the reviewer's point.

---

## [Decision Letter · Decision Letter 1]

20 Sep 2022

PONE-D-22-17093R1The Prevalence of Psychiatric Comorbidities in Adult ADHD Compared With Non-ADHD Populations : A Systematic Literature ReviewPLOS ONE

Dear Dr. Bahk,

Thank you for submitting your manuscript to PLOS ONE. After careful consideration, we feel that it has merit but does not fully meet PLOS ONE’s publication criteria as it currently stands. Therefore, we invite you to submit a revised version of the manuscript that addresses the points raised during the review process.

We look forward to receiving your revised manuscript.

Kind regards,

Soumitra Das

Academic Editor

PLOS ONE

Journal Requirements:

Additional Editor Comments :

Please revise accordingly 

Reviewers' comments:

Reviewer's Responses to Questions

**Comments to the Author**

1. If the authors have adequately addressed your comments raised in a previous round of review and you feel that this manuscript is now acceptable for publication, you may indicate that here to bypass the “Comments to the Author” section, enter your conflict of interest statement in the “Confidential to Editor” section, and submit your "Accept" recommendation.

Reviewer #1: (No Response)

Reviewer #3: All comments have been addressed

2. Is the manuscript technically sound, and do the data support the conclusions?

Reviewer #1: Yes

Reviewer #3: Partly

3. Has the statistical analysis been performed appropriately and rigorously? 

Reviewer #1: Yes

Reviewer #3: N/A

4. Have the authors made all data underlying the findings in their manuscript fully available?

Reviewer #1: Yes

Reviewer #3: Yes

5. Is the manuscript presented in an intelligible fashion and written in standard English?

Reviewer #1: No

Reviewer #3: Yes

6. Review Comments to the Author

Reviewer #1: Thanks for the opportunity to re-review the 'The Prevalence of Psychiatric Comorbidities in Adult ADHD Compared With Non-ADHD Populations : A Systematic Literature Review'. It is clear that the authors were very responsive to all the reviewers' comments, addressed the comments and edited the manuscript accordingly and adequately, or provided detailed justifications otherwise. It results in a massive improvement in the quality of the manuscript. Congratulations!

There are a few follow up concerns/comments for the authors to consider.

1. In my previous comment (Reviewer #1, 4.), the authors responded that

"We appreciate for your suggestion. We updated the searched articles after

22 July 2021. We conducted searches up to August 1, 2022 using the same inclusion &

exclusion criteria for each database. As a result, 2 more studies were included to our review.

We also updated the flow chart, tables and manuscript. Updated numbers and searching

strategy of articles are presented at Supplement 2.

It is great to see that the authors updated the search, which made the findings stay highly up to date. However, this information has not been accurately reflected in the revised manuscript:

As seen in the current revised Abstract: 'Thirty studies published before July 2021 were identified and classified according to diagnosis of other psychiatric disorder in those with ADHD.'

Methods: '...publications regarding the epidemiology and prevalence rate of comorbidities of adult ADHD published from 1 January 1990 to 22 July 2021.'

2. In the limitation section, the authors added the new text

'When our review was evaluated by AMSTAR-2 (85), which evaluates the quality of systematic review studies, our review was rated as low quality review.'

I appreciate that the authors provided a detailed response to address my comment (Reviewer #1, 5) regarding the limitation of not evaluating the risk of bias on each study; and provided a detailed explanation within the manuscript, '...Additionally, we did not assess the risk of bias in each studies. In order to include as many studies as possible to reflect broad and various studies of different countries, populations and comorbid psychiatric disorders, we skipped the risk of bias assessment....'

The authors further evaluated their review based on AMSTAR-2 and concluded that their current study as 'a low quality review'.

According to Shea et al., 2017, a low quality review means "the review has a critical flaw and may not provide an accurate and comprehensive summary of the available studies that address the question of interest". At the moment, without a detailed elaboration on what AMSTAR-2 evaluates, what it actually means 'as a low quality review', what the 'critical flaw' is in the current review, and how it 'may not provide an accurate and comprehensive summary of the available studies that address the question of interest'; it may confuse the readers and devalues the merit of the current review unnecessarily. As the authors have already adequately justified why the risk of bias assessment was not conducted, I suggest the authors to re-consider the necessity and value of including this additional information.

3. A thorough proofread and editing is required for the updated text to enhance readability:

eg.

i) In the method section, data collection,

'Microsoft Excel was used to develop a data extraction sheet, and the whole included full-text

articles were reviewed by two researchers (WSC, YSW) in duplicate who also

conducted the initial data search and study selection process.'

While the authors explained the process clearly in their response (Reviewer #1, 2.):

'...We wanted to mean that both two researchers reviewed all article which was selected by them in any order...' However, the expression in the manuscript still seems to be unclear, especially after putting the words 'in duplicate'. Suggest to reword.

Ii) Study selection section, 'a full-text review was performed to define eligibility of each studies.'; discussion section, ''...Additionally, we did not assess the risk of bias in each studies....'

Should it be each study?

Iii) In discussion, 'The most frequent comorbid psychiatric disorder in the ADHD group was SUD, it’s prevalence ranging from between 2.3% and 41.2% in the general population and between 10.0% and 82.9% in the clinical population',

Should it be ranging from xx to xx?

iv) In conclusion section updated text, 'Which could help for differential diagnosis and providing earlier treatment to adult ADHD individual.'

It seems to be an dependable clause rather than a complete sentence.

Reviewer #3: Thank you again for the opportunity to go through the revised submission. The authors have attempted to respond to most queries. I appreciate that. However, they seem to have introduced many grammatical errors in doing so. Also, when I commented on risk of bias assessment, it is important to understand that risk of bias assessment does not only apply to randomized trials, instead there are other tools such as NIH tool for quality check in observation studies or GRADE for overall quality check. I however appreciate the authors effort in using the AMSTAR checklist for quality check.

Finally, I did not find authors’ response to the following comment though it appears that they have tried to keep these in limitations in response to comments from reviewer 1.

Other important points:

Systematic review protocols are required to be registered before you start the review process to avoid duplications and assist in transparency of the work. There are various options for registration such as PROSPERO, the Registry of Systematic Reviews/Meta-Analyses in Research Registry etc.

In the manuscript submitted, I do not see the registration number. Could the authors include the registration number and the registry where the protocol has been registered.

The authors have not described if they have used quality control for the studies included. In other words, risk of bias assessment is one of the most important steps in any systematic review and metanalysis. Also, who did the quality assessment of the studies included?

Overall, the review seems satisfactory to me. All the best.

7. PLOS authors have the option to publish the peer review history of their article (what does this mean?). If published, this will include your full peer review and any attached files.

Reviewer #1: No

Reviewer #3: **Yes: **Saraswati Dhungana

---

## [Author Response · Author response to Decision Letter 1]

28 Sep 2022

Responses to the Reviewer’s comment

29 September, 2022

Dear reviewers and editorial staffs of PLOS ONE

We are sincerely grateful for your detailed review and consideration of our manuscript, “The Prevalence of Psychiatric Comorbidities in Adult ADHD Compared With Non-ADHD Populations : A Systematic Literature Review”, control number PONE-D-22-17093. Through the comments of first revision process, we could improve the quality of our research. We have additionally revised the manuscript according to the Reviewer’s comments. We performed additional English proofreading, and revised the Reference section in the style required by PLOS ONE (Vancouver style). We acknowledge that the quality of our manuscript was much improved after revising with the scrutinizing efforts of the reviewers and editors. 

Point-by-point responses to the reviewers’ comments are provided in ‘Response to Reviewers’ and revised text in manuscript is highlighted and underlined in red at ‘Revised Manuscript with Track Changes’. Point-by-point responses to the reviewers’ comments are provided below.

Reviewer #1

1. In my previous comment (Reviewer #1, 4.), the authors responded that

"We appreciate for your suggestion. We updated the searched articles after 22 July 2021. We conducted searches up to August 1, 2022 using the same inclusion & exclusion criteria for each database. As a result, 2 more studies were included to our review. We also updated the flow chart, tables and manuscript. Updated numbers and searching strategy of articles are presented at Supplement 2.” It is great to see that the authors updated the search, which made the findings stay highly up to date. However, this information has not been accurately reflected in the revised manuscript: As seen in the current revised Abstract: 'Thirty studies published before July 2021 were identified and classified according to diagnosis of other psychiatric disorder in those with ADHD.' Methods: '...publications regarding the epidemiology and prevalence rate of comorbidities of adult ADHD published from 1 January 1990 to 22 July 2021.'

Author’s response : We appreciate the reviewer’s comment. We have confirmed the error in our manuscript. We reflected the updated information and revised the Abstract, Method, Results, and Discussion section of our manuscript. Thank you for the opportunity to correct the mistake according to the reviewer's point. The revised text is shown below :

Abstract

…Thirty-two studies published before August 2022 were identified and classified according to diagnosis of other psychiatric disorder in those with ADHD.

Method

…We searched electronic libraries of PubMed, EMBASE, PsycINFO, PsycNET, and Google Scholar for publications regarding the epidemiology and prevalence rate of comorbidities of adult ADHD published from 1 January 1990 to 1 August 2022.

Results

…Thus, 32 studies of comparing the prevalence rates of comorbid psychiatric disorders between ADHD and non-ADHD adult subjects were selected for systematic review.

…Of the 32 studies comparing the prevalence of comorbid psychiatric disorder between subjects with and without adult ADHD, according to our classification criteria, 11 studies involved general populations, 18 studies included psychiatric populations, and three studies focused on incarcerated populations.

…The most used evaluation tool for adult ADHD was Adult ADHD Self-Report Scale (ASRS) (25, 26) , which was used in 12 studies. Five studies used the ASRS alone to evaluate ADHD in adults, and the rest of the studies used more than one tool to evaluate ADHD.

Discussion

…In our systematic review, we included 32 studies conducted over 16 years dealing with the prevalence of psychiatric comorbidities between adults with and without ADHD. 

2. In the limitation section, the authors added the new text

'When our review was evaluated by AMSTAR-2 (85), which evaluates the quality of systematic review studies, our review was rated as low quality review.'

I appreciate that the authors provided a detailed response to address my comment (Reviewer #1, 5) regarding the limitation of not evaluating the risk of bias on each study; and provided a detailed explanation within the manuscript, '...Additionally, we did not assess the risk of bias in each studies. In order to include as many studies as possible to reflect broad and various studies of different countries, populations and comorbid psychiatric disorders, we skipped the risk of bias assessment....' The authors further evaluated their review based on AMSTAR-2 and concluded that their current study as 'a low quality review'. According to Shea et al., 2017, a low quality review means "the review has a critical flaw and may not provide an accurate and comprehensive summary of the available studies that address the question of interest". At the moment, without a detailed elaboration on what AMSTAR-2 evaluates, what it actually means 'as a low quality review', what the 'critical flaw' is in the current review, and how it 'may not provide an accurate and comprehensive summary of the available studies that address the question of interest'; it may confuse the readers and devalues the merit of the current review unnecessarily. As the authors have already adequately justified why the risk of bias assessment was not conducted, I suggest the authors to re-consider the necessity and value of including this additional information.

Author’s response : We appreciate the reviewer’s thoughtful suggestion. We agree with the reviewer’s opinion that current revised manuscript can confuse the readers. After discussion with other authors, we have decided to delete the sentence which could devalue the meaning of current study. This will clarify the meaning of our study. Thank you for the reviewer’s comment.

3. A thorough proofread and editing is required for the updated text to enhance readability: eg.

i) In the method section, data collection,

'Microsoft Excel was used to develop a data extraction sheet, and the whole included full-text

articles were reviewed by two researchers (WSC, YSW) in duplicate who also

conducted the initial data search and study selection process.'

While the authors explained the process clearly in their response (Reviewer #1, 2.):

'...We wanted to mean that both two researchers reviewed all article which was selected by them in any order...' However, the expression in the manuscript still seems to be unclear, especially after putting the words 'in duplicate'. Suggest to reword.

Ii) Study selection section, 'a full-text review was performed to define eligibility of each studies.'; discussion section, ''...Additionally, we did not assess the risk of bias in each studies....'

Should it be each study?

Iii) In discussion, 'The most frequent comorbid psychiatric disorder in the ADHD group was SUD, it’s prevalence ranging from between 2.3% and 41.2% in the general population and between 10.0% and 82.9% in the clinical population',

Should it be ranging from xx to xx?

iv) In conclusion section updated text, 'Which could help for differential diagnosis and providing earlier treatment to adult ADHD individual.'

It seems to be an dependable clause rather than a complete sentence.

Author’s response : We appreciate the reviewer’s comment. We confirmed the part mentioned by the reviewer and revised the sentence in manuscript. Also, we received additional English proofreading for the sentences updated to the manuscript. We have attached a confirmation of English proofreading. Thank you for the reviewer’s comment. The revised text is shown below :

Study selection

…The initial inclusion/exclusion review was based on titles and abstracts ; if the relevance of the article was unclear, a full-text review was performed to determine the eligibility of each study.

Data collection process

…Microsoft Excel was used to develop a data extraction spreadsheet, and all included full-text articles were reviewed by both researchers (WSC, YSW), who also conducted the initial data search and study selection process.

Discussion

…The most frequent comorbid psychiatric disorder in the ADHD group was SUD, it’s prevalence ranging from 2.3% and 41.2% in the general population and between 10.0% and 82.9% in the clinical population;

Conclusion

…This standardization will aid in differential diagnosis and allow provision of earlier treatment in adult ADHD.

Reviewer # 3

Reviewer #3: Thank you again for the opportunity to go through the revised submission. The authors have attempted to respond to most queries. I appreciate that. However, they seem to have introduced many grammatical errors in doing so. Also, when I commented on risk of bias assessment, it is important to understand that risk of bias assessment does not only apply to randomized trials, instead there are other tools such as NIH tool for quality check in observation studies or GRADE for overall quality check. I however appreciate the authors effort in using the AMSTAR checklist for quality check.

Finally, I did not find authors’ response to the following comment though it appears that they have tried to keep these in limitations in response to comments from reviewer 1.

Other important points:

Systematic review protocols are required to be registered before you start the review process to avoid duplications and assist in transparency of the work. There are various options for registration such as PROSPERO, the Registry of Systematic Reviews/Meta-Analyses in Research Registry etc.

In the manuscript submitted, I do not see the registration number. Could the authors include the registration number and the registry where the protocol has been registered.

The authors have not described if they have used quality control for the studies included. In other words, risk of bias assessment is one of the most important steps in any systematic review and meta-analysis. Also, who did the quality assessment of the studies included?

Overall, the review seems satisfactory to me. All the best.

Author’s response : We appreciate the reviewer’s thoughtful comments. We performed additional English proofreading for sentences added after first revision. We have attached a confirmation of English proofreading. Also, it was our mistake for not responding to reviewer’s comments about the registration and risk of bias assessment. We apologize for causing the reviewer any inconvenience. The following paragraph is answer for reviewer’s comment : 

As answered to Reviewer #1 in the previous Response to Reviewers, we also agree that the registration process and risk of bias assessment are important for systematic review. However, due to our study was systematic literature review scoping for summarizing and providing overview about prevalence of comorbid psychiatric disorders in adult ADHD, we could not register our study in PROSPERO. This point is important limitation of this study, so we presented this points in limitation section. Also, in our review, we did not performed meta-analysis or statistical synthesis with extracted data, but we summarized and re-arranged the prevalence of comorbid psychiatric disorders in multiple studies comparing ADHD and non-ADHD adults to examine the trend of the difference of each comorbid psychiatric disorders. In order to include as many studies as possible to reflect broad and various studies of different countries, populations and comorbid psychiatric disorders, we did not assess the quality or risk of bias of each included study. This is also why we presented our study as a systematic literature review. We presented this points in Limitation section of our manuscript. We hope that this could be the explanation for not assessing risk of bias of each study which could affect the limited results of our review.

---

## [Decision Letter · Decision Letter 2]

19 Oct 2022

PONE-D-22-17093R2The Prevalence of Psychiatric Comorbidities in Adult ADHD Compared With Non-ADHD Populations : A Systematic Literature ReviewPLOS ONE

Dear Dr. Bahk,

Thank you for submitting your manuscript to PLOS ONE. After careful consideration, we feel that it is almost acceptable for publication in PLOS ONE. However,  some minor corrections in English usage are necessary before acceptance. Please submit a revised version of the manuscript after proofreading the whole manuscript and making necessary corrections including the below mentioned points.

We look forward to receiving your revised manuscript.

Kind regards,

Daimei Sasayama, M.D., Ph.D.

Academic Editor

PLOS ONE

Journal Requirements:

Additional Editor Comments:

Two minor errors were pointed out by one of the reviewer. 

In addition, as shown below, I noticed some other sections that need to be corrected.

Study search and data sources section:

 "The methodology of the present review followed by Preferred Reporting Items for Systematic Reviews and Meta-Analysis (PRISMA)." should be corrected to "The methodology of the present review followed the Preferred Reporting Items for Systematic Reviews and Meta-Analysis (PRISMA)."

"...based on exclusion criteria of non-relevant articles (e.g., did not focus on adult patients or did not include psychiatric comorbidity data), non-English articles, full text not available, abstract-only papers, articles that were not peer-reviewed." should be "...based on exclusion criteria of non-relevant articles (e.g., did not focus on adult patients or did not include psychiatric comorbidity data), non-English articles, full text not available, abstract-only papers, and articles that were not peer-reviewed."

Results section:

"After the duplicates excluded, an additional 1121 articles were excluded by..." should be "After the duplicates were excluded, an additional 1121 articles were excluded by..."

"Thus, 32 studies of comparing the prevalence rates of comorbid psychiatric disorders between ADHD and non-ADHD adult subjects were selected for systematic review." should be "Thus, 32 studies comparing the prevalence rates of comorbid psychiatric disorders between ADHD and non-ADHD adult subjects were selected for systematic review."

Disussion section:

"This might because the interest in adult ADHD had increased based the results of

longitudinal follow-up studies on children and adolescents with ADHD." should be "This might be because the interest in adult ADHD had increased based on the results of longitudinal follow-up studies on children and adolescents with ADHD." 

"The most frequent comorbid psychiatric disorder in the ADHD group was SUD, it's prevalence ranging from..." should be "The most frequent comorbid psychiatric disorder in the ADHD group was SUD, its prevalence ranging from..." or "The most frequent comorbid psychiatric disorder in the ADHD group was SUD, the prevalence of which ranging from..."

"was estimated at 4.45% to 35.3% in the general population at 7.4% to 80.8% in the clinical population." should be "was estimated at 4.45% to 35.3% in the general population and 7.4% to 80.8% in the clinical population."

"...as identified by low hedonic responsibility" should be "...as identified by low hedonic responsivity"

"The prevalence rate in the ADHD group was estimated to rang e from 4.3% to 47.1%" should be "The prevalence rate in the ADHD group was estimated to range from 4.3% to 47.1%"

"Only one study of a clinical population dealing with psychiatric outpatients in the US (34), showed a higher prevalence of adjustment disorder in the non-ADHD group." should be "Only one study of a clinical population dealing with psychiatric outpatients in the US (34) showed a higher prevalence of adjustment disorder in the non-ADHD group."

Table 3:

Capitalize "C" in "childhood-identified ADHD with M.I.N.I (+)" 

Reviewers' comments:

Reviewer's Responses to Questions

**Comments to the Author**

1. If the authors have adequately addressed your comments raised in a previous round of review and you feel that this manuscript is now acceptable for publication, you may indicate that here to bypass the “Comments to the Author” section, enter your conflict of interest statement in the “Confidential to Editor” section, and submit your "Accept" recommendation.

Reviewer #1: All comments have been addressed

Reviewer #3: All comments have been addressed

2. Is the manuscript technically sound, and do the data support the conclusions?

Reviewer #1: Yes

Reviewer #3: Yes

3. Has the statistical analysis been performed appropriately and rigorously? 

Reviewer #1: N/A

Reviewer #3: N/A

4. Have the authors made all data underlying the findings in their manuscript fully available?

Reviewer #1: Yes

Reviewer #3: Yes

5. Is the manuscript presented in an intelligible fashion and written in standard English?

Reviewer #1: Yes

Reviewer #3: Yes

6. Review Comments to the Author

Reviewer #1: Thanks for the opportunity to re-review the manuscript. The authors have been very responsive to all the comments and concerns. The language expression has been greatly improved. The information of the updated search has been shown accurately throughout the manuscript.

I recommend the current version of the manuscript is at the publication standard in principle, but there are two very minor issues of the revised text that would like to draw the authors' attention.

Under study selection: 'We included all types of except a systematic review or meta-analysis defined by title and method.' ?all types of "research"

Under limitation: 'study.-To include as many studies as possible to reflect broad and various studies of different countries,'. There is a '-' in front of the 'To' in the clean version of copy. I believe that it may be picked up at the final publication stage anyway.

Other than these, it is a very high quality of work!

Reviewer #3: I have no further comments to the authors. All queries have been reviewed by the authors and I am satisfied with their final reviison.

7. PLOS authors have the option to publish the peer review history of their article (what does this mean?). If published, this will include your full peer review and any attached files.

Reviewer #1: No

Reviewer #3: **Yes: **Saraswati Dhungana

---

## [Author Response · Author response to Decision Letter 2]

20 Oct 2022

Responses to the Reviewer’s comment

20 October, 2022

Dear reviewers and editorial staffs of PLOS ONE

We are sincerely grateful for your thorough consideration and scrutiny of our manuscript, “The Prevalence of Psychiatric Comorbidities in Adult ADHD Compared With Non-ADHD Populations : A Systematic Literature Review”, control number PONE-D-22-17093R2. Our study is a systematic literature review of recent studies dealing with adults with ADHD which compares the prevalence of comorbid psychiatric disorders between ADHD and non-ADHD populations. Dealing with the comprehensive prevalence of comorbid psychiatric disorders in adults with ADHD, our study will provide broad scope for understanding of comorbid psychiatric disorders in adults with ADHD to readers.

Through the careful comments made by the editor and reviewers, we were able to improve the quality of our manuscript. We have revised the manuscript according to the editor and reviewer’s suggestions. We conducted an additional proofreading, and corrected some errors in the manuscript including Tables. We hope that our revised manuscript will be considered and accepted for publication in the PLOS ONE in shortly. We acknowledge that the scientific and clinical quality of our manuscript was improved by the scrutinizing efforts of the reviewers and editors.

Point-by-point responses to the editor and reviewers’ comments are provided below.

For the editor’s additional comments 

In addition, as shown below, I noticed some other sections that need to be corrected.

Study search and data sources section:

 "The methodology of the present review followed by Preferred Reporting Items for Systematic Reviews and Meta-Analysis (PRISMA)." should be corrected to "The methodology of the present review followed the Preferred Reporting Items for Systematic Reviews and Meta-Analysis (PRISMA)."

"...based on exclusion criteria of non-relevant articles (e.g., did not focus on adult patients or did not include psychiatric comorbidity data), non-English articles, full text not available, abstract-only papers, articles that were not peer-reviewed." should be "...based on exclusion criteria of non-relevant articles (e.g., did not focus on adult patients or did not include psychiatric comorbidity data), non-English articles, full text not available, abstract-only papers, and articles that were not peer-reviewed."

Results section:

"After the duplicates excluded, an additional 1121 articles were excluded by..." should be "After the duplicates were excluded, an additional 1121 articles were excluded by..."

"Thus, 32 studies of comparing the prevalence rates of comorbid psychiatric disorders between ADHD and non-ADHD adult subjects were selected for systematic review." should be "Thus, 32 studies comparing the prevalence rates of comorbid psychiatric disorders between ADHD and non-ADHD adult subjects were selected for systematic review."

Discussion section:

"This might because the interest in adult ADHD had increased based the results of

longitudinal follow-up studies on children and adolescents with ADHD." should be "This might be because the interest in adult ADHD had increased based on the results of longitudinal follow-up studies on children and adolescents with ADHD." 

"The most frequent comorbid psychiatric disorder in the ADHD group was SUD, it's prevalence ranging from..." should be "The most frequent comorbid psychiatric disorder in the ADHD group was SUD, its prevalence ranging from..." or "The most frequent comorbid psychiatric disorder in the ADHD group was SUD, the prevalence of which ranging from..."

"was estimated at 4.45% to 35.3% in the general population at 7.4% to 80.8% in the clinical population." should be "was estimated at 4.45% to 35.3% in the general population and 7.4% to 80.8% in the clinical population."

"...as identified by low hedonic responsibility" should be "...as identified by low hedonic responsivity"

"The prevalence rate in the ADHD group was estimated to rang e from 4.3% to 47.1%" should be "The prevalence rate in the ADHD group was estimated to range from 4.3% to 47.1%"

"Only one study of a clinical population dealing with psychiatric outpatients in the US (34), showed a higher prevalence of adjustment disorder in the non-ADHD group." should be "Only one study of a clinical population dealing with psychiatric outpatients in the US (34) showed a higher prevalence of adjustment disorder in the non-ADHD group."

Table 3:

Capitalize "C" in "childhood-identified ADHD with M.I.N.I (+)"

Author’s response : We appreciate the editor’s thoughtful comments. We revised the mentioned text in our manuscript. And also, we fixed some grammatical errors in the table. Thank you for the opportunity to reconfirm the manuscript and the editor’s efforts to improve the quality of our manuscript.

Reviewer #1

1. Thanks for the opportunity to re-review the manuscript. The authors have been very responsive to all the comments and concerns. The language expression has been greatly improved. The information of the updated search has been shown accurately throughout the manuscript.

I recommend the current version of the manuscript is at the publication standard in principle, but there are two very minor issues of the revised text that would like to draw the authors' attention.

Under study selection: 'We included all types of except a systematic review or meta-analysis defined by title and method.' ?all types of "research"

Under limitation: 'study.-To include as many studies as possible to reflect broad and various studies of different countries,'. There is a '-' in front of the 'To' in the clean version of copy. I believe that it may be picked up at the final publication stage anyway.

Other than these, it is a very high quality of work!

Author’s response : We appreciate the reviewer’s thoughtful comments. We revised the mentioned text in our manuscript. Thank you for the reviewer's compliment and efforts to improve the quality of our manuscript.

Reviewer # 3

I have no further comments to the authors. All queries have been reviewed by the authors and I am satisfied with their final revision.

Author’s response : We appreciate the reviewer’s thoughtful comments. Thank you for your efforts to improve the quality of our manuscript.

---

## [Editor Report · Decision Letter 3]

24 Oct 2022

The Prevalence of Psychiatric Comorbidities in Adult ADHD Compared With Non-ADHD Populations : A Systematic Literature Review

PONE-D-22-17093R3

Dear Dr. Bahk,

We’re pleased to inform you that your manuscript has been judged scientifically suitable for publication and will be formally accepted for publication once it meets all outstanding technical requirements.

Kind regards,

Daimei Sasayama, M.D., Ph.D.

Academic Editor

PLOS ONE
---

## [Editor Report · Acceptance letter]

28 Oct 2022

PONE-D-22-17093R3 

The Prevalence of Psychiatric Comorbidities in Adult ADHD Compared With Non-ADHD Populations : A Systematic Literature Review 

Dear Dr. Bahk:

I'm pleased to inform you that your manuscript has been deemed suitable for publication in PLOS ONE. Congratulations! Your manuscript is now with our production department. 

Kind regards, 

on behalf of

Dr. Daimei Sasayama 

Academic Editor

PLOS ONE